# Evaluating Disentanglement of Structured Representations

**Raphaël Dang-Nhu**

## Abstract

We introduce the first metric for evaluating disentanglement at individual hierarchy levels of a structured latent representation. Applied to object-centric generative models, this offers a systematic, unified approach to evaluating (i) object separation between latent slots (ii) disentanglement of object properties inside individual slots (iii) disentanglement of intrinsic and extrinsic object properties. We theoretically show that for structured representations, our framework gives stronger guarantees of selecting a good model than previous disentanglement metrics. Experimentally, we demonstrate that viewing object compositionality as a disentanglement problem addresses several issues with prior visual metrics of object separation. As a core technical component, we present the first representation probing algorithm handling slot permutation invariance.

## 1 Introduction

A salient challenge in generative modeling is the ability to decompose the representation of images and scenes into distinct objects that are represented separately and then combined together. Indeed, the capacity to reason about objects and their relations is a central aspect of human intelligence (Spelke et al., 1992), which can be used in conjunction with graph neural networks or symbolic solvers to enable relational reasoning. For instance, (Wang et al., 2018) exploit robot body structure to obtain competitive performance and transferability when learning to walk or swim in the Gym environments. (Yi et al., 2019) perform state-of-the-art visual reasoning by using an object based representation in combination with a symbolic model. In the last years, several models have been proposed to learn compositional representations in an unsupervised fashion: TAGGER (Greff et al., 2016), NEM (Greff et al., 2017), R-NEM (Van Steenkiste et al., 2018), MONet (Burgess et al., 2019), IODINE (Greff et al., 2019), GENESIS (Engelcke et al., 2019), Slot Attention (Locatello et al., 2020), MulMON (Li et al., 2020), SPACE (Lin et al., 2020). They jointly learn to represent individual objects and to segment the image into meaningful components, the latter often being called "perceptual grouping". These models share a number of common principles: (i) Splitting the latent representation into several *slots* that are meant to contain the representation of an individual object. (ii) Inside each slot, encoding information about both object position and appearance. (iii) Maintaining a symmetry between slots in order to respect the permutation invariance of objects composition. These mechanisms are intuitively illustrated in Figure 1.

To compare and select models, it is indispensable to have robust disentanglement metrics. At the level of individual factors of variations, a representation is said to be disentangled when information about the different factors is separated between different latent dimensions (Bengio et al., 2013; Locatello et al., 2019). At object-level, disentanglement measures the degree of object separation between slots. However, all existing metrics (Higgins et al., 2016; Chen et al., 2018; Ridgeway and Mozer, 2018; Eastwood and Williams, 2018; Kumar et al., 2017) are limited to the individual case, which disregards representation structure. To cite Kim and Mnih (2018) about the FactorVAE metric:

> The definition of disentanglement we use [...] is clearly a simplistic one. It does not allow correlations among the factors or hierarchies over them. Thus this definition seems more suited to synthetic data with independent factors of variation than to most realistic datasets.

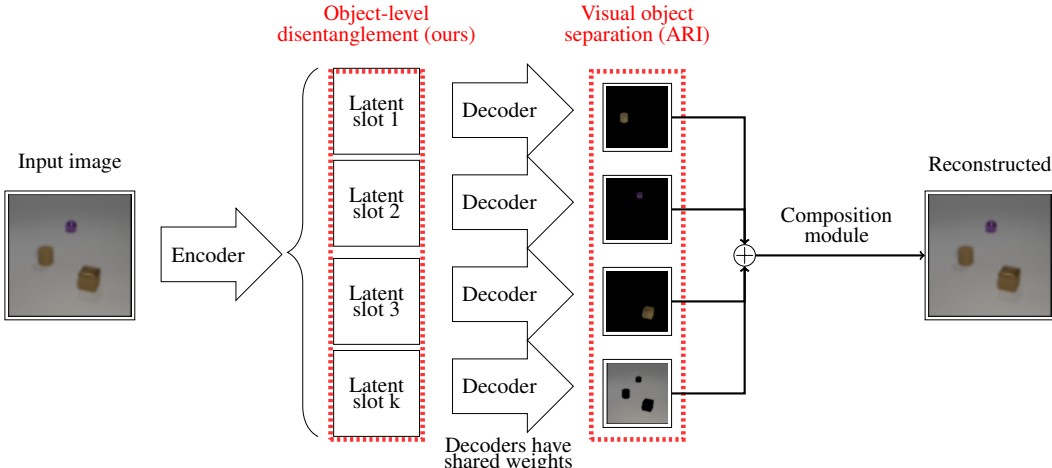

Figure 1: A compositional latent representation is composed of several *slots*. Each slot generates a part of the image. Then, the different parts are composed together. Pixel-level metrics measure object separation between slots at visual level, while our framework operates purely in latent space.

As a result, prior work has restricted to measuring the degree of object separation via pixel-level segmentation metrics. Most considered is the Adjusted Rand (ARI) Index (Rand, 1971; Greff et al., 2019), where image segmentation is viewed as a cluster assignment for pixels. Other metrics such as Segmentation Covering (mSC) (Arbelaez et al., 2010) have been introduced to penalize over-segmentation of objects. A fundamental limitation is that they do not evaluate directly the quality of the representation, but instead consider a visual proxy of object separation. This results in problematic dependence on the quality of the inferred segmentation masks, a problem first identified by Greff et al. (2019) for IODINE, and confirmed in our experimental study.

To address these limitations, we introduce the first metric for evaluating disentanglement at individual hierarchy levels of a structured latent representation. Applied to object-centric generative models, this offers a systematic, unified approach to evaluating (i) object separation between latent slots (ii) disentanglement of object properties inside individual slots (iii) disentanglement of intrinsic and extrinsic object properties. We theoretically show that our framework gives stronger guarantees of representation quality than previous disentanglement metrics. Thus, it can safely substitute to them. We experimentally demonstrate the applicability of our metric to three architectures: MONet, GENESIS and IODINE. The results confirm issues with pixel-level segmentation metrics, and offer valuable insight about the representation learned by these models. Finally, as a core technical component, we present the first representation probing algorithm handling slot permutation invariance.

## 2 BACKGROUND

### 2.1 DISENTANGLEMENT CRITERIA

There exists an extensive literature discussing notions of disentanglement, accounting for all the different definitions is outside the scope of this paper. We chose to focus on the three criteria formalized by Eastwood and Williams (2018), which stand out because of their clarity and simplicity. *Disentanglement* is the degree to which a representation separates the underlying factors of variation, with each latent variable capturing at most one generative factor. *Completeness* is the degree to which each underlying factor is captured by a single latent variable. *Informativeness* is the amount of information that a representation captures about the underlying factors of variation. Similarly to prior work, the word *disentanglement* is also used as a generic term that simultaneously refers to these three criteria. It should be clear depending on the context whether it is meant as general or specific.

## 2.2 The DCI framework

For brevity reasons, we only describe the DCI metrics of Eastwood and Williams (2018), which are most closely related to our work. The supplementary material provides a comprehensive overview of alternative metrics. Consider a dataset $\mathcal{X}$ composed of $n$ observations $x^1, \ldots, x^n$, which we assume are generated by combining $F$ underlying factors of variation. The value of the different factors for observation $x^l$ are denoted $v_1^l, \ldots, v_F^l$. Suppose we have learned a representation $z = (z_1, \ldots, z_L)$ from this dataset. The DCI metric is based on the affinity matrix $R = (R_{i,j})$, where $R_{i,j}$ measures the relative importance of latent $z_i$ in predicting the value of factor $v_j$. Supposing appropriate normalization for $R$, disentanglement is measured as the weighted average of the matrix' row entropy. Conversely, completeness is measured as the weighted average of column entropy. Informativeness is measured as the normalized error of the predictor used to obtain the matrix $R$.

## 3 Evaluating structured disentanglement

3.1 contains a high level description of the goals steering our framework. Our metric is formally described in 3.2. In 3.3, we present theoretical results showing that our framework can safely substitute to DCI since it provides stronger guarantees that the selected model is correctly disentangled.

### 3.1 Presentation of the framework

Prior work has focused on measuring disentanglement at the level of individual factors of variation and latent dimensions (that we will refer to as *global* or *unstructured* disentanglement). In the case of compositional representations with several objects slots, additional properties are desirable:

1. **Object-level disentanglement:** Objects and latent slots should have a one-to-one matching. That is, changing one object should lead to changes in a single slot, and vice-versa.

2. **Slot disentanglement:** Inside a given slot, properties of the object must be disentangled.

3. **Slot symmetry:** The latent dimension responsible for a given factor (e.g., color) should be invariant across slots. This means that all slots should have the same inner structure.

Our structured disentanglement metric allows to evaluate all these criteria within a single unified framework. Similarly to DCI, it is based on the affinity matrix $(R_{\hat{\tau}, \tau})$, where $R_{\hat{\tau}, \tau}$ measures the relative importance of latent $z_{\hat{\tau}}$ in predicting the value of factor $v_\tau$. The key novelty compared to DCI is that we propose to measure disentanglement with respect to arbitrary **projections** of the matrix $R$. Intuitively, projections correspond to a marginalization operation that selects a subset of hierarchy levels and discards the others. The coefficients $R_{\hat{\tau}, \tau}$ that are projected together are summed to create group affinities. Figure 2 gives an intuitive illustration of this process on a toy example.

With object-centric representations, projecting at the object/slot level allows to study the relation of objects and slots without taking their internal structure into consideration. Ultimately this permits to evaluate our object-level disentanglement criterion. Projecting at property level allows to study the internal slot representation independently of the object, for evaluation of both internal slot disentanglement and slot symmetry, with a single metric. The identity projection conserving all levels measures flat (DCI) disentanglement. This generalizes to arbitrary hierarchies in the representation, such as disentanglement of position and appearance, or intrinsic and extrinsic object properties.

### 3.2 Mathematical definition

To formalize our framework most conveniently, we propose to view the affinity matrix $R = (R_{\hat{\tau}, \tau})$ as a joint random variable $(X, Y)$, where $X$ is a random latent dimension, and $Y$ a random factor. Supposing that $R$ is normalized to have sum one, this means $\mathbb{P}[X = \hat{\tau}, Y = \tau] = R_{\hat{\tau}, \tau}$. This point of view, which is only implicit in prior work, allows to formalize the projections of $R$ as a coupled marginalization operation $(\rho(X), \rho(Y))$, where $\rho(\alpha_1, \ldots, \alpha_h) = (\alpha_{e_1}, \ldots, \alpha_{e_l})$ selects a subset of hierarchy levels. This yields a concise and elegant mathematical framework, where we can build on standard information-theoretic identities to derive theoretical properties. In the following, $H_U(A|B)$ denotes the conditional entropy of $A$ with respect to $B$ (detailed definitions in the supplementary).

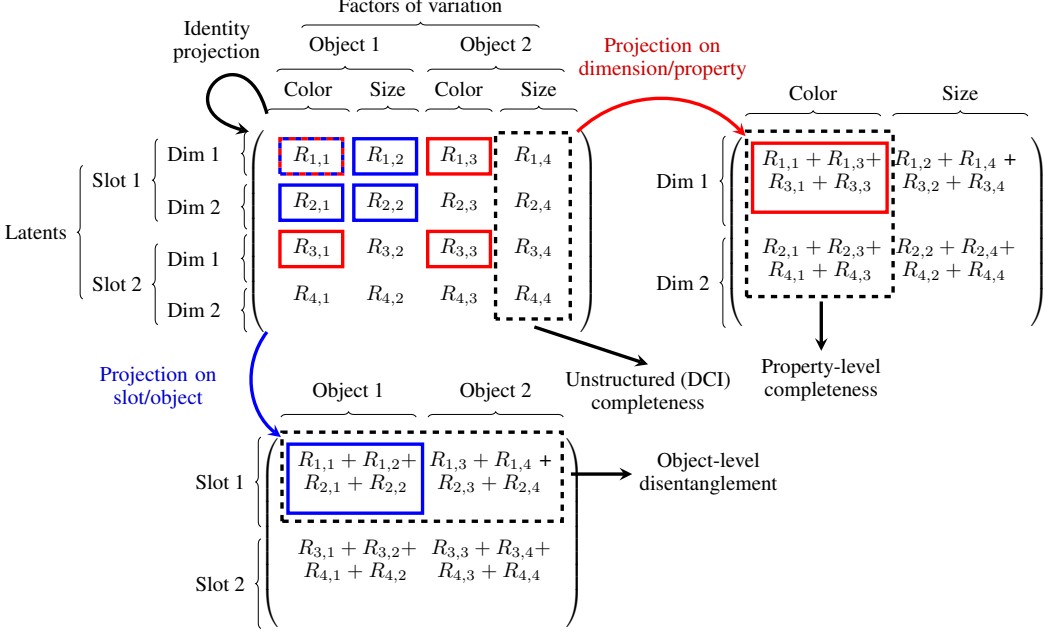

Figure 2: Overview of projections on a toy example of object-centric representation. The affinity scores $R$ are marginalized according to the selected hierarchy levels. In projected space, row entropy measures disentanglement, while column entropy measures completeness. The measure of entropy involves a renormalization by row (for completeness) or column (for disentanglement). The affinity scores $R$ are computed in an all-pairs manner. A mapping can be obtained (e.g. from object to slot) by looking for the maximum of each column.

Our disentanglement criteria are the following:

**Completeness** with respect to projection $\rho$ measures how well the projected factors are captured by a coherent group of latents, by measuring column entropy in the projected space. It is measured as

$$C(\rho) = 1 - H_U\left(\rho(X)|\rho(Y)\right),$$

where $U = |\rho(\hat{\mathcal{T}})|$ is the number of groups of latents in the projection.

**Disentanglement** with respect to projection $\rho$ measures to what extent a group of latent variables influences a coherent subset of factors, by measuring projected row entropy. It is measured as

$$D(\rho) = 1 - H_V\left(\rho(Y)|\rho(X)\right),$$

where $V = |\rho(\mathcal{T})|$ is the number of groups of factors in the projection.

**Informativeness** does not depend on the projection. It is defined as the normalized error of a low-capacity model $f$ that learns to predict the factor values $v$ from the latents $z$, i.e.

$$I = \frac{\|f(z) - v\|_2}{\|v\|_2}.$$

The changing log bases $U$ and $V$ aim at ensuring normalization between $0$ and $1$.

### 3.3 Establishing trust in our framework: a theoretical analysis

Our disentanglement $D(\rho)$ and completeness $C(\rho)$ metrics depend on the subset of hierarchy levels contained in the projection $\rho$. We theoretically analyze the influence of the projection choice. It is especially interesting to study the behavior when distinct subsets of hierarchy levels are combined. Our key results are the following: Theorem 1 shows that our framework contains DCI as a special case, when the identity projection is chosen. Theorem 2 shows that the metrics associated to the identity projection can be decomposed in terms of one dimensional projections along a single hierarchy level. Together, these results formally show that **one dimensional projections provide a sound substitute for prior unstructured metrics**. This is very useful to build trust in our framework.

**Theorem 1** (Relation with DCI). *The DCI metrics of Eastwood and Williams (2018) are a special case of our framework, with the identity projection conserving all hierarchy levels.*

Theorem 2 uses the intuitive notion of decomposition of a projection. The projection $\rho$ is said to be decomposed into $\rho_1, \ldots \rho_k$ if the set of hierarchy levels selected by $\rho$ is the disjunct union of the set of levels selected by $\rho_1, \ldots \rho_k$. In the case of object-centric latent representations, the identity projection $\rho_{\text{id}}$ that keeps all hierarchy levels $\{\text{object}, \text{property}\}$ can be decomposed as $\rho_{\text{object}}$ considering only $\{\text{object}\}$ and $\rho_{\text{property}}$ considering only $\{\text{property}\}$.

**Theorem 2** (Decomposition of a projection). *Consider a decomposition of the projection $\rho$ (with $L$ latent groups) into disjunct projections $\rho_1, \ldots \rho_k$ (with respectively $L^1, \ldots, L^k$ latent groups). The following lower bound for the joint completeness (resp. disentanglement) holds*

$$1 - k + \sum_{s=1}^{k} C(\rho_s) \leq 1 - \sum_{s=1}^{k} \frac{1 - C(\rho_s)}{\log_{L^s}(L)} \leq C(\rho).$$

Suppose that all one dimensional projections verify $C(\rho_s) \geq 1 - \epsilon$, where $\epsilon \geq 0$. We obtain the lower bound $C(\rho_{\text{id}}) \geq (1 - k) + k(1 - \epsilon) = 1 - k\epsilon$ for the identity projection. For object-centric representations, this implies that when object-level completeness and property completeness are both perfect (that is, $\epsilon = 0$), then DCI completeness is also perfect. The same works for disentanglement. The supplementary materials contains detailed proofs, a matching upper bound, as well as an explicit formula in the special case $k = 2$.

## 4 Permutation invariant representation probing

*Representation probing* aims at quantifying the information present in each latent dimension, to obtain the relative importance $R_{\hat{\tau}, \tau}$ of latent $z_{\hat{\tau}}$ in predicting factor $v_\tau$. Traditional probing methods either use regression feature importance or mutual information to obtain $R$ (see the supplementary). However, these techniques do not account for the permutation invariance of object-centric representations, in which slot reordering leaves the generated image unchanged. To address this, we propose a novel formulation as a permutation invariant feature importance problem. As a core technical contribution of this paper, we present an efficient EM-like algorithm to solve this task in a tractable way. Traditional representation probing fits a regressor $f$ to predict the factors $v$ from latents $z$, solving

$$\arg\min_f \sum_{j=1}^{n} \left\| v^j - f(z^j) \right\|_2^2.$$

Then, the matrix $R$ is extracted from the feature importances of $f$. In contrast, our formulation of permutation invariant representation probing jointly optimizes on permutations $(\pi_1, \ldots, \pi_n)$ over groups of latent dimensions (for instance slots), allowing to account for slot permutation invariance of the representation, thus solving

$$\arg\min_{(\pi_1, \ldots, \pi_n), f} \sum_{j=1}^{n} \left\| v^j - f(\pi_j(z^j)) \right\|_2^2.$$

To address the combinatorial explosion created by the joint optimization, we chose an EM-like approach and iteratively optimize $f$ and the permutations $(\pi_1, \ldots, \pi_n)$, while keeping the other fixed (Algorithm 1). This method yields satisfying approximate solutions provided a good initialization (See Figure 5 and the supplementary).

---

**Algorithm 1** Permutation-invariant representation probing

---

**Input:** Latent codes $z^1, \ldots, z^n$ and factor values $v^1, \ldots, v^n$ for $n$ input images
**for** $i = 1$ **to** n_iters **do**
  *# M STEP*
  Fit predictor $f_i$ to features $z^1, \ldots, z^n$ and targets $v^1, \ldots, v^n$.
  *# E STEP*
  **for** $j = 1$ **to** $n$ **do**
    $\pi_{\min} \leftarrow \arg\min_{\pi} \left\| v^j - f_i(\pi(z^j)) \right\|_2^2$ *# ($\pi$ ranges on all slot permutations)*
    $z^j \leftarrow \pi_{\min}(z^j)$
Fit $f_{\mathrm{final}}$ to features $z^1, \ldots, z^n$ and targets $v^1, \ldots, v^n$.
Obtain feature importances from $f_{\mathrm{final}}$.

---

## 5 EXPERIMENTAL EVALUATION

**Models** We compare three different architectures with public Pytorch implementations: MONet, GENESIS and IODINE, which we believe to be representative of object-centric representations. We evaluate two variants of MONet: the first one, denoted as "MONet (Att.)" follows the original design in Burgess et al. (2019) and uses the masks inferred by the attention network in the reconstruction. The second, denoted as "MONet (Dec.)" is a variant that instead uses the decoded masks. For all models, we perform an ablation of the disentanglement regularization in the training loss. Ablated models are trained with pure reconstruction loss, denoted as (R). All our final feature importances are obtained with random forests.

**Datasets** We evaluate all models on CLEVR6 (Johnson et al., 2017) and Multi-dSprites (Matthey et al., 2017; Burgess et al., 2019), with the exception of IODINE that we restricted to Multi-dSprites for computational reasons, as CLEVR6 requires a week on 8 V100 GPUs per training. These datasets are the most common benchmarks for learning object-centric latent representations. Both contain images generated compositionally. Multi-dSprites has 2D shapes on a variable color background, while CLEVR6 is composed of realistically rendered 3D scenes. We slightly modified CLEVR6 to ensure that all objects are visible in the cropped image. Table 3 contains our experimental results at both object and property-level, compared to pixel-level segmentation metrics (ARI and mSC). We highlight some of these numbers in Figure 3. Figure 4 shows the different projections of the affinity matrix $R$ as Hinton diagrams. Figure 5 visualizes the slot permutation learned by our permutation invariant probing algorithm.

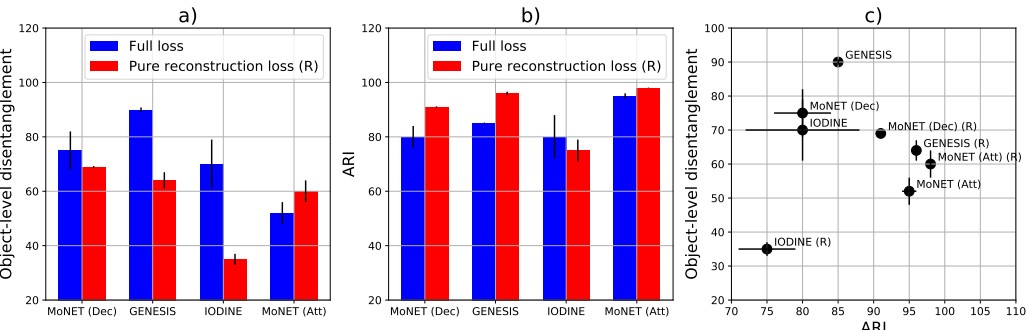

Figure 3: a) Ablation of disentanglement regularization tends to *decrease* disentanglement at object-level (Multi-dSprites), b) Ablation of disentanglement regularization tends to *increase* pixel-level segmentation (measured with ARI on Multi-dSprites). c) Object-level disentanglement vs. ARI for different models trained on Multi-dSprites. We observe a negative correlation between both metrics.

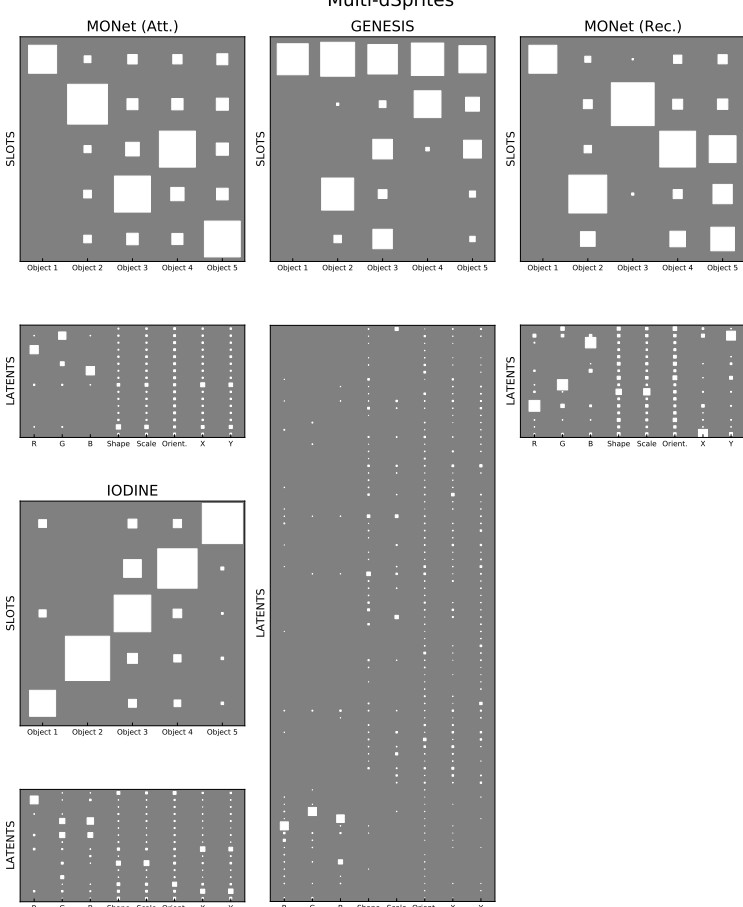

Figure 4: Projections of the affinity matrix on Multi-dSprites for IODINE, GENESIS and the two variants of MONet, as Hinton diagrams: The white area is proportional to coefficient value.

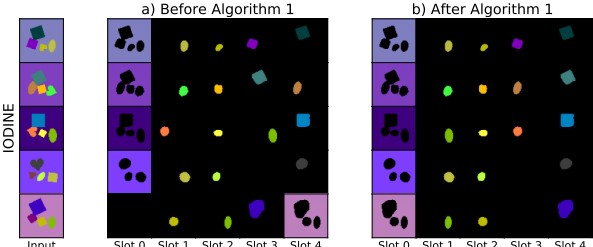

Figure 5: Effect of EM probing (Alg. 1) on slot ordering for a group of similar inputs. Each row corresponds to the decomposition of a given input between slots. a) is without EM probing, b) with EM-probing. The decomposition is compared to a reference decomposition (on the first row). We observe that EM-probing assigns objects to slot in a way that is always consistent to the reference input. That is, similar objects are matched to similar slots.

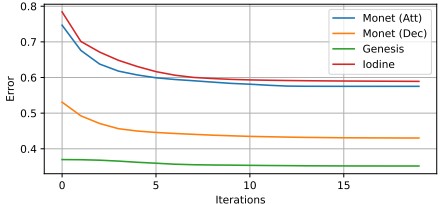

Figure 6: Convergence of the EM probing algorithm, for the different models (Multi-dSprites). Values can not be directly compared with Table 2 as the final predictor has higher capacity.

Table 1: Comparison of the permutation obtained with EM probing vs obtained with IoU matching (multi-dSprites, averaged across models). We report (i) percentage of exactly matching permutations (ii) average mismatch between both permutations.

|  | % Exactly matching | # Average mismatch |
| --- | --- | --- |
| Measured | 62 % | 1.08 |
| Best possible | 100 % | 0 |
| Worst possible | 0 % | 5 |
| Chance level | 0.83 % | 4 |

## 5.1 Relation to pixel-level segmentation metrics.

Our object-level metrics have the same goal as visual metrics such as ARI and mSC: quantifying object separation between slots. Therefore, we observe a general correlation of all metrics, with low values at initialization and improvement throughout training. The main difference is that ARI and mSC evaluate separation at visual level, while our framework purely operates in latent space, measuring the repartition of information (see Figure 1). Consequently, ARI and mSC tend to favor sharp object segmentation masks, while our framework is focused on easy extraction of information from the latent space. This fundamental difference leads to specific regimes of negative correlation. Figure 3 c) shows that ARI and our object-level metric give very different rankings of trained model performance for Multi-dSprites, suggesting that our framework addresses the dependence of ARI on sharp segmentation masks identified by Greff et al. (2019). This is particularly visible for IODINE, which achieves good disentanglement despite its low ARI.

## 5.2 Influence of the disentanglement regularization

Figure 3 a) and b) shows an ablation study of the disentanglement regularization. The ablated models are trained with pure reconstruction loss, without regularization of the latent space. For our disentanglement metrics, we observe a consistent negative impact. This is consistent with observations for unstructured representations (Eastwood and Williams, 2018; Locatello et al., 2019). On the contrary, the ablation tends to improve pixel-level segmentation. This would imply that visual object separation is only caused by architectural inductive biases, which clarifies the impact of disentanglement regularization in prior work.

## 5.3 Insights about the different models

Qualitatively, visual inspection of the different projections of matrix $R$ in Figures 4 and 11 shows a near one-to-one mapping between slots and objects, except some redundancy for the background slot, which is excluded from our metrics. Our quantitative results in Table 3 confirm a near perfect object disentanglement of up to 90% for Multi-dSprites. For CLEVR6 we consider the value of $50\%$ to be decent, because the logarithmic scale of the conditional entropy tends to create a strong shift towards 0. To give a simple reference, if each object is equally influenced by 2 slots among 6, the expected value is $1 - \log_6(2) = 57\%$. Table 3 additionally shows that (i) the robust performance, of GENESIS supports the generalization of its separate mask encoding process and GECO optimizer. (ii) that IODINE is able to get close to the performance of MONet, despite a less stable training process for IODINE which resulted in an outlier with bad performance. (iii) MONet (Att.) obtains better visual separation metrics, while MONet (Dec.) has better disentanglement. This is because using the decoded masks in MONet (Dec.) increases the pressure to accurately encode the masks in the latent representation. On the contrary, MONet (Att.) does not use the decoded masks for reconstruction, and thus does not need to encode them very accurately.

## 5.4 Extension to other hierarchy levels

To demonstrate generalization to more than two hierarchy levels, we evaluate disentanglement of intrinsic and extrinsic object properties. This specifically applies to GENESIS, for which each slot is divided in a mask latent and component latent. Intuitively, intrinsic properties (such as shape and color) are related to the nature of the object, whereas extrinsic properties (such as position and

Table 2: Projection along a third hierarchy level (GENESIS, MdSprites).

|  | Extrinsic | Intrinsic |
|---|---|---|
| Mask latent | 0.2 | 0.32 |
| Component latent | 0 | 0.48 |

orientation) are contextual to the scene. The projection along this third hierarchy level is given in Table 2, for GENESIS trained on Multi-dSprites. Numbers show that extrinsic properties are successfully captured by the mask latent. However, intrinsic properties are not satisfyingly separated. This is because information about object shape is contained both in the segmentation mask and in the generated image of the object. We believe that recent architectures (Nguyen-Phuoc et al., 2020; Ehrhardt et al., 2020) performing object composition in 3D scene space might solve this problem.

Table 3: Experimental results (stderr. over three seeds, values in %). Models with (R) have no disentanglement regularization. ARI/mSC is slightly higher (resp. lower) than prior work for Multi-dSprites (resp. CLEVR6,) because of minor differences in the data generation process. At object-level, we observe that disentanglement and completeness match closely, but this is not the case at property-level, due to the asymmetry in the number of latent factors and slot dimensions.

| | | OBJECT-LEVEL | | PROPERTY-LEVEL | | | | |
| | MODEL | DIS. ($\uparrow$) | COMP. ($\uparrow$) | DIS.($\uparrow$) | COMP.($\uparrow$) | INF.($\downarrow$) | ARI($\uparrow$) | MSC ($\uparrow$) |
|---|---|---|---|---|---|---|---|---|
| MDSPR. | MONET (ATT.) | $52 \pm 4$ | $52 \pm 4$ | $39 \pm 6$ | $60 \pm 5$ | $45 \pm 3$ | $95 \pm 1$ | $\mathbf{83 \pm 0.3}$ |
| | MONET (ATT.) (R) | $60 \pm 4$ | $60 \pm 4$ | $29 \pm 3$ | $52 \pm 1$ | $36 \pm 2$ | $\mathbf{98 \pm 0.1}$ | $86 \pm 4$ |
| | MONET (DEC.) | $75 \pm 7$ | $75 \pm 7$ | $59 \pm 6$ | $\mathbf{74 \pm 5}$ | $26 \pm 6$ | $80 \pm 4$ | $68 \pm 2$ |
| | MONET (DEC.) (R) | $69 \pm 0.3$ | $69 \pm 0.2$ | $35 \pm 1$ | $55 \pm 1$ | $32 \pm 0.8$ | $91 \pm 0.2$ | $69 \pm 2$ |
| | GENESIS | $\mathbf{90 \pm 0.8}$ | $\mathbf{91 \pm 0.6}$ | $\mathbf{72 \pm 0.5}$ | $60 \pm 0.4$ | $\mathbf{24 \pm 0.4}$ | $85 \pm 0$ | $70 \pm 0.3$ |
| | GENESIS (R) | $64 \pm 3$ | $65 \pm 3$ | $65 \pm 1$ | $42 \pm 1$ | $30 \pm 0.5$ | $96 \pm 0.6$ | $\mathbf{84 \pm 0.7}$ |
| | IODINE | $70 \pm 9$ | $72 \pm 10$ | $37 \pm 5$ | $62 \pm 4$ | $37 \pm 6$ | $80 \pm 8$ | $71 \pm 3$ |
| | IODINE (R) | $35 \pm 2$ | $36 \pm 1$ | $17 \pm 2$ | $46 \pm 1$ | $61 \pm 2$ | $75 \pm 4$ | $65 \pm 1$ |
| CLEVR6 | MONET (ATT.) | $\mathbf{46 \pm 5}$ | $\mathbf{48 \pm 5}$ | $22 \pm 4$ | $\mathbf{51 \pm 3}$ | $55 \pm 5$ | $\mathbf{93 \pm 0.4}$ | $68 \pm 6$ |
| | MONET (ATT.) (R) | $40 \pm 9$ | $41 \pm 9$ | $13 \pm 5$ | $44 \pm 3$ | $61 \pm 8$ | $\mathbf{93 \pm 0.4}$ | $65 \pm 5$ |
| | MONET (DEC.) | $\mathbf{47 \pm 6}$ | $\mathbf{48 \pm 6}$ | $21 \pm 6$ | $50 \pm 4$ | $55 \pm 5$ | $90 \pm 2$ | $65 \pm 2$ |
| | MONET (DEC.) (R) | $40 \pm 1$ | $41 \pm 0.2$ | $13 \pm 0.6$ | $44 \pm 0.1$ | $60 \pm 0.6$ | $92 \pm 0.5$ | $\mathbf{69 \pm 1}$ |
| | GENESIS | $\mathbf{50 \pm 2}$ | $\mathbf{52 \pm 2}$ | $\mathbf{47 \pm 2}$ | $39 \pm 2$ | $\mathbf{47 \pm 0.9}$ | $91 \pm 0.4$ | $65 \pm 3$ |
| | GENESIS (R) | $43 \pm 1$ | $45 \pm 2$ | $24 \pm 2$ | $21 \pm 2$ | $65 \pm 1$ | $92 \pm 0.2$ | $60 \pm 3$ |

## 5.5 RE-EXAMINING THE MODEL SELECTION PROCESS

These divergences between visual metrics and our framework lead us to reconsider prior model selection processes, which were primarily centered on ARI. To give three striking examples (i) the recent Slot Attention (Locatello et al., 2020) architecture does not use disentanglement regularization. In light of our ablation study, we believe that this is potentially harmful for disentanglement (Section 5.2). (ii) Burgess et al. (2019) and Engelcke et al. (2019) chose to privilege MOnet (Att.) which obtains lower disentanglement that MONet (Dec.) (Section 5.3) (iii) most prior work chose a 2D segmentation mask approach that does not satisfyingly disentangle intrinsic and extrinsic object properties (5.4). We believe that all related existing experimental studies would benefit from the perspective offered by our framework.

## 6 RELATED WORK

**Structured disentanglement metrics.** Most similar to our work is the slot compactness metric of Racah and Chandar (2020), also based on aggregation of feature importances to measure object separation between slots. However, it only operates at one hierarchy level and does not handle slot permutation invariance, which is essential to obtain meaningful results. Also related is the hierarchical disentanglement benchmark of Ross and Doshi-Velez (2021), whose ability to learn the hierarchy levels in the representation, is remarkable, but is unfortunately limited in its applicability to toy datasets. Finally, Esmaeili et al. (2019) present a structured variational loss encouraging disentanglement at group-level. There is a high-level connection with our work, but the objective is ultimately different. Esmaeili et al. (2019) evaluate their structured variational loss with unstructured disentanglement metrics.

## 7 CONCLUSION

Our framework for evaluating disentanglement of structured latent representations addresses a number of issues with prior visual segmentation metrics. We hope that it will be helpful for validation and selection of future object-centric representations. Besides, we took great care in not making any domain specific assumption, and believe that the principles discussed here apply to any kind of structured generative modelling.

## Acknowledgements

This work was granted access to the HPC resources of IDRIS under the allocation 2020-AD011012138 made by GENCI. We would like to thank Frederik Benzing, Kalina Petrova, Asier Mujika, and Wouter Tonnon for helpful discussions. This work constitutes the public version of Raphaël Dang-Nhu's Master Thesis at ETH Zürich.

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

# A  ADDITIONAL RELATED WORK

## A.1  OBJECT-CENTRIC REPRESENTATION LEARNING

Many models have been developed to perform unsupervised perceptual grouping and learn compositional representations. A first line of work (TAGGER (Greff et al., 2016), NEM (Greff et al., 2017), R-NEM (Van Steenkiste et al., 2018)) has focused on adapting traditional Expectation Maximization (EM) (Dempster et al., 1977) methods for data clustering to a differentiable neural setting. Nonetheless, despite strong theoretical foundations, these models do not scale to more complex multi-object datasets such as Multi-dSprites (Burgess et al., 2019) and CLEVR (Johnson et al., 2017). More recent efforts (MONet (Burgess et al., 2019), IODINE (Greff et al., 2019), GENESIS (Engelcke et al., 2019)), Slot Attention (Locatello et al., 2020)) have focused on application to these datasets.

These architectures differ by how exactly they segment the image between the different slots of the representation. MONet is based on a recurrent attention network that repeatedly separates parts of the image until it is totally decomposed. GENESIS was designed to address a key limitation of MONet, namely that it does not learn a latent representation for the segmentation, which prevents principled sampling of novel scenes. With GENESIS, segmentation masks are separately encoded, and an autoregressive prior on the mask latents is enforced. IODINE uses a different strategy building upon the framework of iterative amortized inference, where an initial arbitrary guess for the posterior is progressively refined. However, this iterative process is very computationally expensive. Slot Attention, as its name indicates, introduces an attention based iterative encoder that is much more computationally efficient than Iodine. The SPACE model (Lin et al., 2020) combines the strength of spatial attention with scene mixture models to further improve applicability. Finally, MulMON (Li et al., 2020) extends object-centric representations to a setting with multi-view supervision.

In a parallel line of work, object-compositional generationis believed to be a promising inductive bias for GAN-like models (van Steenkiste et al., 2018). Obtaining good representations and producing realistic samples are connected challenges, which both require to develop suitable decoder architectures. Very recent work has been focusing on integrating object compositionality into the GAN framework (Nguyen-Phuoc et al., 2020; Ehrhardt et al., 2020; Niemeyer and Geiger, 2020). The question of disentanglement is also present in this context (Nguyen-Phuoc et al., 2019), but it remains a secondary objective.

## A.2  TRADITIONAL DISENTANGLEMENT METRICS

Several metrics (Zaidi et al., 2020) have been developed to evaluate the quality of learned representations and allow comparison between models. We identify two main categories:

**Classifier-based metrics**  The first group of metrics is based on fixing the value of a factor of variation and generating several samples sharing this value. Intuitively, if the factors of variation are satisfyingly disentangled in the representation, it should be possible to predict which factor was fixed from the different latents. Inside this category, the metric differ in how they exactly identify the factor. BetaVAE (Higgins et al., 2016) uses a linear classifier to predict the factor index, while FactorVAE uses majority vote and takes the empirical variances as input, with the goal of addressing several robustness issues. Despite these implementation differences, Locatello et al. (2019) have empirically observed that both metrics are strongly correlated across a wide range of tasks and models.

**Affinity-based metrics**  The second category of metrics is based on measuring the affinity between each factor of variation and each latent variable, and evaluating how close these are to a one-to-one mapping between factors and variables. Different ways of quantifying affinity have been proposed: the Mutual Information Gap (Chen et al., 2018) and Modularity (Ridgeway and Mozer, 2018) measure mutual information between factors and variables. The SAP score (Kumar et al., 2017) leverages the linear coefficient of determination $R^2$ obtained when regressing the factor from the latent. Finally, the DCI metric is based on measuring regression feature importance, using either Lasso or random forests. These metrics also differ on how they exactly assess the separation of factors in the latent representation. The Mutual Information Gap and the SAP score measure the gap between the two most relevant variables for a factor. Modularity alternatively suggests to quantify the distance to an

ideal affinity template. Finally, the DCI metric uses the entropy of the normalized feature importances as an indicator of entanglement.

In developing our metric, we have chosen to privilege the affinity-based approach for two reasons. First, it does not require the ability to fix one factor while generating samples, contrary to BetaVAE and FactorVAE. Second, the notion of affinity generalizes in a very flexible way to group of factors and latent variables: it is sufficient to sum the scores of all pairs of factors and variables in the groups. We use regression feature importance as a way of measuring affinity. Moreover, we privilege the entropy measure of separation over affinity-gap methods. Indeed, we believe that entropy captures more information about the repartition of information as it takes into account all coefficients rather than just the top two.

### A.3 Permutation invariant learning

Developing algorithms that preserve permutation-invariance is no doubt a major challenge for the machine learning community: similar problematics appear in a variety of applications, ranging from speaker separation in the cocktail-party problem (Yu et al., 2017) to object detection losses (Carion et al., 2020). Compared to the representation probing of IODINE, the Hungarian matching in Slot Attention, and the IoU matching of MulMON (Li et al., 2020), the fundamental novelty or Algorithm 1 is that it obtains the optimal alignment *directly at latent code level (that is, feature level)*, which is agnostic to the input type. In contrast, the alignment in IODINE and MulMON operates at image level using the segmentation mask, and the Hungarian matching in Slot Attention aligns factor predictions and labels. Besides, the matching approaches used in Iodine, Slot Attention and MulMON only identify the most important slot for each target, but the DCI framework requires an importance score for each slot/object pair. For future work, it would be interesting to compare EM-probing with Sinkhorn based approaches for learning latent permutations (e.g. Mena et al. (2018)).

### B Proofs and additional theoretical results

#### B.1 Background in information theory

The description of our metric leverages concepts originating from the field of information-theory Cover (1999). In this section, we recall some central definitions and results that we will use in the rest of the manuscript. In all of the following, $X, Y$ and $Z$ denote three discrete random variables defined on a common probability space. We denote $x_1, \ldots, x_l$ (resp $y_1, \ldots, y_m$ and $z_1, \ldots, z_n$) the potential outcomes of $X$ (resp. $Y$ and $Z$). $K$ is a positive real number. For brevity reasons, we denote $\mathbb{P}[x_i]$ for $\mathbb{P}[X = x_i]$, an similarly for $Y$ and $Z$ and the joint random variables.

**Definition 1.** *The entropy of $X$ in base $K$ quantifies the amount of uncertainty in the potential outcomes of $X$. It is defined as*

$$H_K(X) = -\mathbb{E}[\log_K(\mathbb{P}(X))] = -\sum_{i=1}^{l} \mathbb{P}[x_i] \log_K(\mathbb{P}[x_i]).$$

**Definition 2.** *The conditional entropy of $Y$ given $X$ in base $K$ quantifies the amount of information needed to describe the outcome of $Y$ given than the value of $X$ is known. It is defined as*

$$H_K(Y|X) = -\sum_{i,j} \mathbb{P}[x_i, y_j] \log_K(\mathbb{P}[y_j|x_i]).$$

**Definition 3.** *The mutual information of $X$ and $Y$ in base $K$ is a measure of the amount of information obtained about one of the two variables by observing the other. It is defined as*

$$I_K(X;Y) = \sum_{i,j} \mathbb{P}[x_i, y_j] \log_K \left( \frac{\mathbb{P}[x_i, y_j]}{\mathbb{P}[x_i] \cdot \mathbb{P}[y_j]} \right).$$

**Definition 4.** *The conditional mutual information $I_K(X;Y|Z)$ is, in base $K$, the expected value of the mutual information of $X$ and $Y$ conditioned on the value of $Z$. Formally,*

$$I_K(X;Y|Z) = \sum_{i,j,k} \mathbb{P}[x_i, y_j, z_k] \log_K \left( \frac{\mathbb{P}[x_i, y_j, z_k]}{\mathbb{P}[x_i|z_k]\mathbb{P}[y_j|z_k]} \right)$$

The following Lemmas are standard results of information theory. As these are well-known, we did not provide a proof here and we refer to textbooks such as Cover (1999).

**Lemma 1** (Change of base). *Let $K^1$ be a positive real number. The change of base formula for entropy is*

$$H_{K^1}(X) = \frac{H_K(X)}{\log_{K^1} K}.$$

*Similarly, we have for conditional entropy that*

$$H_{K^1}(Y|X) = \frac{H_K(Y|X)}{\log_{K^1} K}.$$

**Lemma 2** (Subadditivity of entropy). *Let $X_1, \ldots, X_n$ be $n$ arbitrary discrete random variables. We have*

$$H_K(X_1, \ldots, X_n) \leq \sum_{i=1}^{n} H_K(X_i).$$

*The same holds for conditional entropy, i.e.*

$$H_K(X_1, \ldots, X_n|Y) \leq \sum_{i=1}^{n} H_K(X_i|Y).$$

**Lemma 3** (Nonnegativity of mutual information). *The following holds*

$$H_K(X) - H_K(X|Y) = I_K(X;Y) \geq 0.$$

*Conditioned on a third variable, this generalizes to*

$$H_K(X|Z) - H_K(X|Y, Z) = I_K(X;Y|Z) \geq 0.$$

**Lemma 4** (Relation of joint entropy to individual entropies). *Let $X_1, \ldots, X_n$ be $n$ arbitrary discrete random variables. We have*

$$H_K(X_1, \ldots, X_n) \geq \max_{i=1}^{n} H_K(X_i).$$

*The same holds for conditional entropy, i.e.*

$$H_K(X_1, \ldots, X_n|Y) \geq \max_{i=1}^{n} H_K(X_i|Y).$$

**Lemma 5** (Joint conditional entropy). *The following holds*

$$H_K(X, Y|Z) = H_K(X|Z) + H_K(Y|Z) - I_K(X;Y|Z).$$

## B.2 FORMALIZATION OF THE STRUCTURED SETTING

The specificity of our setting compared to the DCI metric is that the structured organization of latent variables and factors of variations can not be accurately described by scalar indexes. We propose to account for this structured organization via tuple indexes, in which each element of the tuple is responsible for a hierarchy level. In general, the structure over factors of variation expresses the goal of metric (e.g. measuring object separation), while the structure over latent dimensions depends on the model architecture.

Formally, both structures can be defined as *relations*. Consider $n$ attributes $A_1, \ldots, A_n$ representing levels of hierarchy. Each attribute is associated with a set of possible values (a domain) for factors $\mathrm{dom}_F(A_i)$ and for latents $\mathrm{dom}_L(A_i)$. These domains constrain the possible values for the tuple indexes, such that the set of tuples $\mathcal{T}$ describing the factors of variation is a subset of $\prod_{i=1}^{n} \mathrm{dom}_F(A_i)$, and the set of tuples $\hat{\mathcal{T}}$ describing the latent dimensions is a subset of $\prod_{i=1}^{n} \mathrm{dom}_L(A_i)$. Thus, the factor values for sample $i$ can be denoted $(v_\tau^i)_{\tau \in \mathcal{T}}$, and the representation $z^i$ can be written $z^i = (z_{\hat{\tau}}^i)_{\hat{\tau} \in \hat{\mathcal{T}}}$. We also denote $|\mathcal{T}| = F$ and $|\hat{\mathcal{T}}| = L$.

**Toy example** we consider a data generating process with two objects having two properties each. The first attribute $A_1$ is responsible for the object hierarchy level, while $A_2$ is responsible for the property level. The structure over factors is formally defined as

$$\mathcal{T} = \{(\text{object 1}, \text{color}), (\text{object 1}, \text{size}),$$
$$(\text{object 2}, \text{color}), (\text{object 2}, \text{size})\},$$

Supposing that there are 2 slots with two dimensions each, the structure over latent dimensions is defined as

$$\hat{\mathcal{T}} = \{(\text{slot 1}, \text{dim 1}), (\text{slot 1}, \text{dim 2}),$$
$$(\text{slot 2}, \text{dim 1}), (\text{slot 2}, \text{dim 2})\}$$

### B.3 Proofs of main paper

**Theorem 1.** *With the total projection $i = \{1, \ldots, n\}$, our metric captures the disentanglement and completeness metrics of the DCI framework (Eastwood and Williams, 2018).*

*Proof.* We will only present the proof for the completeness metric, as the situation for the disentanglement metric is exactly symmetric, when switching $X$ and $Y$ and rows and columns. With the total projection $i = \{1, \ldots, n\}$, we have $\rho_i(Y) = Y$, $\rho_i(X) = X$, $U = |\rho_i(\hat{\mathcal{T}})| = |\hat{\mathcal{T}}| = L$ and $V = |\rho_i(\mathcal{T})| = |\mathcal{T}| = F$.

Therefore, completeness with respect to the total projection is defined as

$$C(i) = 1 - H_L(X|Y).$$

According to Definition 2, we have

$$H_L(X|Y) = -\sum_{\tau \in \mathcal{T}, \hat{\tau} \in \hat{\mathcal{T}}} \mathbb{P}[X = \hat{\tau}, Y = \tau] \log_L(\mathbb{P}[X = \hat{\tau}|Y = \tau])$$
$$= -\sum_{\tau \in \mathcal{T}} \mathbb{P}[Y = \tau] \sum_{\hat{\tau} \in \hat{\mathcal{T}}} \mathbb{P}[\hat{\tau}|\tau] \log_L(\mathbb{P}[\hat{\tau}|\tau]).$$

Now, let us observe that the conditional probability $\mathbb{P}[\hat{\tau}|\tau]$ is exactly the term $\tilde{P}_{\hat{\tau}, \tau}$ of the DCI framework that denotes the "probability" of latent $\hat{\tau}$ being important to predict factor $\tau$. Consequently, the conditional entropy can be rewritten as follows

$$H_L(X|Y) = -\sum_{\tau \in \mathcal{T}} \mathbb{P}[Y = \tau] \sum_{\hat{\tau} \in \hat{\mathcal{T}}} \tilde{P}_{\hat{\tau}, \tau} \log_L(\tilde{P}_{\hat{\tau}, \tau}) = \sum_{\tau \in \mathcal{T}} \mathbb{P}[Y = \tau] H_L(\tilde{P}_{\cdot, \tau}).$$

This yields

$$C(i) = 1 - H_L(X|Y) = \sum_{\tau \in \mathcal{T}} \mathbb{P}[Y = \tau](1 - H_L(\tilde{P}_{\cdot, \tau})) = \sum_{\tau \in \mathcal{T}} \mathbb{P}[Y = \tau] C_\tau.$$

We observe that $C_\tau = 1 - H_L(\tilde{P}_{\cdot, \tau})$ is exactly the completeness score in capturing factor $\tau$ defined in the DCI framework. Besides, $\mathbb{P}[Y = \tau]$ can be rewritten as

$$\mathbb{P}[Y = \tau] = \sum_{\hat{\tau} \in \hat{\mathcal{T}}} \mathbb{P}[Y = \tau, X = \hat{\tau}] = \frac{\sum_{\hat{\tau} \in \hat{\mathcal{T}}} R_{\hat{\tau}, \tau}}{\sum_{i \in \hat{\mathcal{T}}, j \in \mathcal{T}} R_{i, j}},$$

which is exactly the relative generative factor importance used by Eastwood and Williams to construct a weighted average expressing overall completeness. This indicates that our probabilistic view of the affinity matrix and metric naturally captures all components of the DCI framework, including the final weighted average step. □

For the next theorem, we formally define the meaning of a decomposition of projections in Definition 5. Note that the formalism is slightly different as our original notations were simplified for the main paper. Despite differences in notation, union and decomposition of projections are totally equivalent.

**Definition 5** (Union of projections). *Consider $k$ disjunct projections $i^1 = \{e_1^1, \ldots, e_{l_1}^1\}, \ldots, i^k = \{e_1^k, \ldots, e_{l_k}^k\}$ of the generative model. The union of these projections is defined as*

$$i = \bigcup_{s=1}^{k} i^s.$$

*It is a projection of size $l_1 + \ldots + l_k$.*

**Theorem 2.** *[Lower bound] Consider $k$ disjunct projections $i^1, \ldots, i^k$ of the relations. Let us suppose that $i^1, \ldots, i^k$ have respectively $L^1, \ldots, L^k$ groups of latents and $F^1, \ldots, F^k$ groups of factors. Moreover, assume that the joint projection $i = \bigcup_{s=1}^{k} i^s$ has $L$ groups of latents and $F$ groups of factors. The following lower bound for the joint completeness holds*

$$1 - k + \sum_{s=1}^{k} C(i^s) \le 1 - \sum_{s=1}^{k} \frac{1 - C(i^s)}{\log_{L^s}(L)} \le C\left(\bigcup_{s=1}^{k} i^s\right).$$

*Similarly, we have for the disentanglement metric*

$$1 - k + \sum_{s=1}^{k} D(i^s) \le 1 - \sum_{s=1}^{k} \frac{1 - D(i^s)}{\log_{F^s}(F)} \le D\left(\bigcup_{s=1}^{k} i^s\right).$$

*Proof.* We only detail the proof for the completeness metric since both cases are exactly symmetric. $C(i)$ is defined as follows

$$C(i) = 1 - H_L\left(\rho_i(X)|\rho_i(Y)\right).$$

Now, let us observe that the joint projection $\rho_i$ and the concatenation of the different projections $\prod_{s=1}^{k} \rho_{i^s}$ are similar up to the permutation of the dimensions that originates from sorting the merged index sequences. This permutation has no influence on the conditional entropy since it is defined as a joint expectation on $\hat{\mathcal{T}} \times \mathcal{T}$. Therefore we have that

$$C(i) = 1 - H_L\left(\prod_{s=1}^{k} \rho_{i^s}(X) \Big| \prod_{t=1}^{k} \rho_{i^t}(Y)\right). \tag{1}$$

According to Lemma 2, we have the following inequality for the joint conditional entropy

$$H_L\left(\prod_{s=1}^{k} \rho_{\boldsymbol{i}^s}(X) \Big| \prod_{t=1}^{k} \rho_{\boldsymbol{i}^t}(Y)\right) \le \sum_{s=1}^{k} H_L\left(\rho_{\boldsymbol{i}^s}(X) \Big| \prod_{t=1}^{k} \rho_{\boldsymbol{i}^t}(Y)\right).$$

According to Lemma 3, we also know that

$$H_L\left(\rho_{i^s}(X) \Big| \prod_{t=1}^{k} \rho_{i^t}(Y)\right) \le H_L\left(\rho_{i^s}(X)|\rho_{i^s}(Y)\right).$$

Applying Lemma 1 to change base, we obtain

$$H_L\left(\rho_{i^s}(X)|\rho_{i^s}(Y)\right) = \frac{H_{L^s}\left(\rho_{i^s}(X)|\rho_{i^s}(Y)\right)}{\log_{L^s}(L)} = \frac{1 - C(i^s)}{\log_{L^s}(L)}.$$

Together with Equation (1), this yields

$$1 - \sum_{s=1}^{k} \frac{1 - C(i^s)}{\log_{L^s}(L)} \le C\left(\bigcup_{s=1}^{k} i^s\right).$$

Since $\log_{L^s}(L) \ge 1$ and $1 - C(i^s) \ge 0$, we finally get

$$1 - k + \sum_{s=1}^{k} C(i^s) \le 1 - \sum_{s=1}^{k} \frac{1 - C(i^s)}{\log_{L^s}(L)},$$

which concludes the proof. $\qquad\square$

## B.4 ADDITIONAL RESULTS

Theorem 3 describes an upper bound for the joint metric attempting to match Theorem 2. However, this upper bound comes in a weaker form which is not totally controllable. Intuitively, this is due to the fact that our metrics convey strictly more information that the unstructured DCI framework.

**Theorem 3.** *[Upper bound] Consider $k$ disjunct projections $i^1, \ldots, i^k$ of the relations. Let us suppose that $i^1, \ldots, i^k$ have respectively $L^1, \ldots, L^k$ groups of latents and $F^1, \ldots, F^k$ groups of factors. Moreover, assume that the joint projection $i = \bigcup_{s=1}^{k} i^s$ has $L$ groups of latents and $F$ groups of factors. The following upper bound for the completeness metric holds*

$$C\left(\bigcup_{s=1}^{k} i^s\right) \leq 1 - \max_{1 \leq s \leq k} \left(\frac{1 - C(i^s)}{\log_{L^s} L} - A_s\right),$$

*where*

$$A_s = I_L\left(\rho_{i^s}(X); \rho_{\bigcup_{t \neq s} i^t}(Y) | \rho_{i^s}(Y)\right).$$

*Similarly, we have for the disentanglement metric that*

$$D\left(\bigcup_{s=1}^{k} i^s\right) \leq 1 - \max_{1 \leq s \leq k} \left(\frac{1 - D(i^s)}{\log_{F^s} F} - B_s\right).$$

*where*

$$B_s = I_F\left(\rho_{i^s}(Y); \rho_{\bigcup_{t \neq s} i^t}(X) | \rho_{i^s}(X)\right).$$

*Proof.* We only prove the bound for the completeness metric, as the case of disentanglement is exactly symmetric. $C(i)$ is defined as follows

$$C(i) = 1 - H_L\left(\rho_i(X) | \rho_i(Y)\right).$$

Similar to the previous proof, we observe that the joint projection $\rho_i$ and the concatenation of the different projections $\prod_{s=1}^{k} \rho_{i^s}$ are similar up to the permutation of the dimensions that originates from sorting the merged index sequences. This permutation has no influence on the conditional entropy since it is defined as a joint expectation on $\hat{\mathcal{T}} \times \mathcal{T}$. Therefore we have that

$$C(i) = 1 - H_L\left(\prod_{s=1}^{k} \rho_{i^s}(X) | \prod_{t=1}^{k} \rho_{i^t}(Y)\right). \tag{2}$$

According to Lemma 4, we have the following inequality for the joint conditional entropy

$$\max_{s=1}^{k} H_L\left(\rho_{i^s}(X) | \prod_{t=1}^{k} \rho_{i^t}(Y)\right) \leq H_L\left(\prod_{s=1}^{k} \rho_{i^s}(X) | \prod_{t=1}^{k} \rho_{i^t}(Y)\right)$$

According to Lemma 3, we also know that

$$H_L\left(\rho_{i^s}(X) | \prod_{t=1}^{k} \rho_{i^t}(Y)\right) = H_L\left(\rho_{i^s}(X) | \rho_{i^s}(Y)\right) - A_s,$$

where

$$A_s = I_L\left(\rho_{i^s}(X); \rho_{\bigcup_{t \neq s} i^t}(Y) | \rho_{i^s}(Y)\right).$$

Applying Lemma 1 to change base, we obtain

$$H_L\left(\rho_{i^s}(X) | \prod_{t=1}^{k} \rho_{i^t}(Y)\right) = \frac{1 - C(i^s)}{\log_{L^s} L} - A_s.$$

Together with Equation (2), this yields

$$C(i) \leq 1 - \max_{1 \leq s \leq k} \left(\frac{1 - C(i^s)}{\log_{L^s} L} - A_s\right).$$

$\square$

**Interpretation of the upper bound**   The goal of Theorem 3 is to use the individual projections to obtain an upper bound for the joint disentanglement (resp. completeness). Intuitively, the meaning of the bound

$$C(i) \leq 1 - \max_{1 \leq s \leq k} \left( \frac{1 - C(i^s)}{\log_{L^s} L} - A_s \right).$$

is that the joint completeness can not be better than any of the individual completeness. However, this bound is weaker than Theorem 2 because of two main restrictions. First, it is harder to get rid of the $\log_{L^s}(L)$ term as $L$ is greater than $L^s$. One possibility is to notice that $L \leq L^1 \cdot \ldots \cdot L^k$. If we make the additional assumption that $L^1 \sim \ldots \sim L^k$, then we obtain $\log_{L^s}(L) \leq k$, which gives us

$$C(i) \leq 1 - \frac{1}{k} + \min_{1 \leq s \leq k} \left( \frac{C(i^s)}{k} + A_s \right).$$

This first assumption can be considered reasonable. But more importantly, we notice an additional interaction term $A_s$ in the minimum that is based on mutual information between the different projections. This term can not be removed as it is non-negative. The intuition for this conditional mutual information is that it measures the dependence of the different projections. It is $0$ exactly when $\rho_{i^s}(X)$ and $\rho_{\bigcup_{t \neq s} i^t}(Y)$ are independent conditioned on $\rho_{i^s}(Y)$. Assuming that this is case, we obtain the simplest inequality

$$C(i) \leq 1 - \frac{1}{k} + \min_{1 \leq s \leq k} \frac{C(i^s)}{k}.$$

However, there is absolutely no guarantee that this assumption is valid. To conclude, the upper bound is weaker than Theorem 2 because it depends on the additional terms $A_s$ and $B_s$ which account for the degree of dependence between projections and can not be easily controlled. This can be seen as a confirmation that our metrics convey strictly more information that the unstructured DCI framework.

In the following, we study the specific case where there are only two levels of hierarchy ($k = 2$). In this case, we can actually derive an exact formula rather two matching bounds.

**Theorem 4** (Case k = 2). *Consider 2 disjunct projections $i^1$ and $i^2$, with respectively $L^1$ and $L^2$ groups of latents, and $F^1$, $F^2$ groups of factors. Assume that the joint projection $i = i^1 \cup i^2$ has $L$ groups of latents and $F$ groups of factors. For the completeness metric, the following identity holds*

$$
\begin{aligned}
C(i) = 1 &- \frac{1 - C(i^1)}{\log_{L^1}(L)} - \frac{1 - C(i^2)}{\log_{L^2}(L)} \\
&+ I_L(\rho_{i^1}(X); \rho_{i^1}(Y) | \rho_{i^2}(Y)) \\
&+ I_L(\rho_{i^2}(X); \rho_{i^2}(Y) | \rho_{i^1}(Y)) \\
&+ I_L(\rho_{i^1}(X); \rho_{i^2}(X) | \rho_{i^1}(Y), \rho_{i^2}(Y)).
\end{aligned}
$$

*In a symmetric way, for the disentanglement metric,*

$$
\begin{aligned}
D(i) = 1 &- \frac{1 - D(i^1)}{\log_{F^1}(F)} - \frac{1 - D(i^2)}{\log_{F^2}(F)} \\
&+ I_F(\rho_{i^1}(Y); \rho_{i^1}(X) | \rho_{i^2}(X)) \\
&+ I_F(\rho_{i^2}(Y); \rho_{i^2}(X) | \rho_{i^1}(X)) \\
&+ I_F(\rho_i(Y); \rho_{i^2}(Y) | \rho_{i^1}(X), \rho_{i^2}(X)).
\end{aligned}
$$

*Proof.* Again, we only prove the result for the completeness metric as disentanglement is exactly symmetric. In the case where $k = 2$, the joint completeness is defined as follows

$$C(i) = 1 - H_L\left(\rho_i(X) | \rho_i(Y)\right) = 1 - H_L\left(\rho_{i^1}(X), \rho_{i^2}(X) | \rho_{i^1}(Y), \rho_{i^2}(Y)\right).$$

With Lemma 5, we can rewrite this last expression as

$$
\begin{aligned}
C(i) = 1 &- H_L\left(\rho_{i^1}(X) | \rho_{i^1}(Y), \rho_{i^2}(Y)\right) - H_L\left(\rho_{i^2}(X) | \rho_{i^1}(Y), \rho_{i^2}(Y)\right) + \\
&I_L\left(\rho_{i^1}(X); \rho_{i^2}(X) | \rho_{i^1}(Y), \rho_{i^2}(Y)\right).
\end{aligned}
$$

Applying Lemma 3 twice, we obtain

$$H_L\left(\rho_{i^1}(X) | \rho_{i^1}(Y), \rho_{i^2}(Y)\right) = H_L\left(\rho_{i^1}(X) | \rho_{i^1}(Y)\right) - I_L\left(\rho_{i^1}(X); \rho_{i^1}(Y) | \rho_{i^2}(Y), \rho_{i^2}(Y)\right),$$

and

$$H_L\left(\rho_{i^2}(X)|\rho_{i^1}(Y),\rho_{i^2}(Y)\right) = H_L\left(\rho_{i^2}(X)|\rho_{i^2}(Y)\right) - I_L\left(\rho_{i^2}(X);\rho_{i^2}(Y)|\rho_{i^1}(Y).\rho_{i^2}(Y)\right),$$

Using Lemma 1 to change base for the two conditional entropies gives us the final equation

$$\begin{aligned}
C(i) = 1 &- \frac{1 - C(i^1)}{\log_{L^1}(L)} - \frac{1 - C(i^2)}{\log_{L^2}(L)} \\
&+ I_L(\rho_{i^1}(X); \rho_{i^1}(Y)|\rho_{i^2}(Y)) \\
&+ I_L(\rho_{i^2}(X); \rho_{i^2}(Y)|\rho_{i^1}(Y))+ \\
&+ I_L(\rho_{i^1}(X); \rho_{i^2}(X)|\rho_{i^1}(Y),\rho_{i^2}(Y)),
\end{aligned}$$

which concludes the proof. $\qquad\square$

## C  REPRODUCIBILITY

### C.1  PERMUTATION INVARIANT FEATURE PROBING

Complementary to our permutation invariant probing algorithm, we use a specific algorithm for generation of the evaluation dataset. It consists in dividing the evaluation dataset into groups of inputs with similar factor values. First, this helps for the initialization of our EM-like approach, as the object-to-slot assignment exhibits less variation for similar images. Second, this allows to visually verify that the learned permutation is consistent across inputs (Figures 7, 8 and 9). Finally, it breaks the symmetry between objects, allowing for more meaningful visualizations of the projected latent space. To obtain a global metric, we average the values across the different groups. The method for generating the evaluation dataset is summarized in Algorithm 2. Exact hyperparameters are given below, and an ablation study can be found in Section D.

---

**Algorithm 2** Generation of the evaluation dataset

---

**for** $g = 1$ **to** n_groups **do**
  *# Generate initial factor values for the group*
  **for** $f$ **in** factors **do**
    Sample initial value initial[$f$] for factor $f$
  **for** $i = 1$ **to** n_samples **do**
    *# Sample factor values locally*
    **for** $f$ **in** factors **do**
      Sample value image_i[$f$] near initial[$f$]
      Generate image $i$ from factor values image_i

---

### C.2  MODELS

**MONet**   We train MONet exactly as in Burgess et al. (2019), except that we set $\sigma_{fg} = 0.1$ and $\sigma_{bg} = 0.06$ which was shown to yield better results in (Greff et al., 2019). Besides, we use GECO (Rezende and Viola, 2018) to automatically tune the $\beta$ parameter during training. The reconstruction constraint was manually set to ensure satisfying visual results (-1.78 for Multi-dSprites, -1.75 for CLEVR6). We found it especially important for the beginning of the training to set a minimum value of 1 for $\beta$. Failure to do so resulted in frequent local optima for the representation in which the network was stuck. We used the implementation provided by the authors of Engelcke et al. (2019). Note that contrary to the latter, we do not include $\gamma$ in the GECO framework. We use the value of 0.5 given the authors of MONet. The $\gamma$ parameter controls a mask reconstruction loss, therefore we did not set it to 0 in MONet (R).

**GENESIS**   We follow the training process described in (Engelcke et al., 2019) and use the implementation provided by the authors, under GNU General Public License. CLEVR6 is not considered in this paper: we used the same parameters as for Multi-dSprites and changed the reconstruction objective to $-1.428$.

**IODINE**   We use the parameters described in (Greff et al., 2019), and a third-party implementation.[1] For Multi-dSprites, we had to increase the $\sigma$ parameter from $0.1$ to $0.14$ in order to obtain satisfying results with the slight variations in our dataset.

We train all models for 200 epochs. This corresponds to approximately 300 000 updates on Multi-dSprites and 450 000 updates on CLEVR6. Models were trained with one to four V100 GPUs. The training times ranges from a few hours to 1.5 days.

## C.3   Training datasets

**Multi-dSprites**   For training, we use the dataset and preprocessing code from Engelcke et al. (2019). Note that the datasets contains images composed of 1 to 4 objects, contrary to (Greff et al., 2019) that goes up to 5 objects. This explains the better segmentation values in our setting compared to the latter.

**CLEVR6**   We generate the training dataset similarly as in Greff et al. (2019), with the modification that we add an additional constraint that all objects have to be visible inside the cropped 192x192 image. This modification is important for our metric. It tends to generate slightly denser scenes than in previous work. This might partly explain why we get slightly worse ARI for MONet compared to previous work (0.94 against 0.96). Scenes have at most 6 objects.

## C.4   Evaluation

**Evaluation datasets**   The evaluation datasets are sampled in the same way, except that we restrict to images with 4 objects for Multi-dSprites and images with exactly 6 objects for CLEVR6. We sample 10 local groups (see Algorithm 2) for Multi-dSprites and 5 for CLEVR6. Each group has 5000 samples, with a 4000/500/500 split for fitting, validation and evaluation of the factor predictor. In Table 7, we give the list of factors for each individual object in both datasets, with the possible values for each factor.

**Factor prediction**   For the temporary predictors in Algorithm 1 (inside the loop), we use a linear model with Ridge regularization. For the final predictor, we use a random forest with 10 trees, and a maximum depth of 15. This because the random forest obtains better predictions, while the linear model permits faster iterations. The number of iterations of the loop is set to 100 for Multi-dSprites. For computational reasons, we reduced this number to 20 for CLEVR6. This reduction does not harm performance as the vast majority of permutations happen in the first iterations. Most factors are continuous or ordinal: for these, we encode factor prediction as a regression task. The material factor in CLEVR6 has only two classes and we follow the encoding used in sklearn Ridge Classifier. On both datasets, the shape factor has three classes: we generalize the Ridge Classifier encoding and encode the classes as -1, 0 and -1. We match the classes to these values in order to minimize prediction error.

**Max tree depth**   We tried different max tree depths (5, 10, 15, 20, 25, 30) on a validation set and observed that (i) a value of 15 generally gives better validation performance than 5 and 10 (ii) values > 15 do not significantly improve validation performance and get very slow to fit. This is why we chose to always go with 15. Note that this optimal value might be related to the number of samples used to fit the predictor (currently 4000). Note that the validation was done globally, and not per factor.

**Metric**   In order to filter out noise in the feature importances, we set to 0 to all the relative importance coefficients that are less than $3\%$ of the column maximum in absolute value. In our evaluation of object-level disentanglement, we remove the factors and latent dimensions that are background related. For Multi-dSprites, the background slot is identified as the one with higher importance in predicting background color. For CLEVR6, there is no background color, but the background is always reconstructed by the first slot in the benchmarked models. The global informativeness metric is an unweighted average over all factors.

---

[1] github.com/zhixuan-lin/IODINE, no provided license.

Table 4: Ablation study of Algorithms 1 and 2 for Multi-dSprites. We report mean and std error over three random seeds. All values are in %. 100 is the best possible score and 0 the worst, except for the informativeness metric where 0 is best. All models are trained with full loss.

| | OBJECT-LEVEL | |
| MODEL | DIS. (↑) | COMP. (↑) |
|---|---|---|
| MONET (ATT.) | $52 \pm 4$ | $52 \pm 4$ |
| MONET (ATT.) (WITHOUT ALG 1) | $30 \pm 2$ | $30 \pm 2$ |
| MONET (ATT.) (WITHOUT ALG 2) | $57 \pm 1$ | $57 \pm 1$ |
| MONET (ATT.) (WITHOUT ALG 1 AND 2) | $22 \pm 0.5$ | $22 \pm 0.5$ |
| MONET (REC.) | $75 \pm 7$ | $75 \pm 7$ |
| MONET (REC.) (WITHOUT ALG 1) | $40 \pm 3$ | $40 \pm 4$ |
| MONET (REC.) (WITHOUT ALG 2) | $64 \pm 4$ | $64 \pm 4$ |
| MONET (REC.) (WITHOUT ALG 1 AND 2) | $24 \pm 1$ | $24 \pm 1$ |
| GENESIS | $90 \pm 0.8$ | $91 \pm 0.6$ |
| GENESIS (WITHOUT ALG 1) | $89 \pm 0.1$ | $90 \pm 0.1$ |
| GENESIS (WITHOUT ALG 2) | $61 \pm 0.4$ | $61 \pm 0.4$ |
| GENESIS (WITHOUT ALG 1 AND 2) | $33 \pm 0.3$ | $33 \pm 0.3$ |
| IODINE | $70 \pm 9$ | $72 \pm 10$ |
| IODINE (WITHOUT ALG 1) | $32 \pm 2$ | $33 \pm 2$ |
| IODINE (WITHOUT ALG 2) | $66 \pm 1$ | $66 \pm 1$ |
| IODINE (WITHOUT ALG 1 AND 2) | $26 \pm 0.9$ | $26 \pm 0.9$ |

Table 5: Informativeness and Completeness per object property for all models trained with full loss on Multi-dSprites.

| | COMPLETENESS (↑) | | | | INFORMATIVENESS (↓) | | | |
| | GENESIS | IODINE | MONET (A) | MONET (DEC) | GENESIS | IODINE | MONET (A) | MONET (DEC) |
|---|---|---|---|---|---|---|---|---|
| R CHANNEL | 91 | 71 | 88 | 94 | 9 | 16 | 5 | 5 |
| G CHANNEL | 90 | 68 | 81 | 89 | 10 | 17 | 6 | 7 |
| B CHANNEL | 90 | 77 | 86 | 96 | 9 | 16 | 4 | 5 |
| SHAPE | 30 | 49 | 42 | 50 | 26 | 45 | 59 | 34 |
| SCALE | 32 | 49 | 42 | 55 | 38 | 53 | 66 | 41 |
| ORIENTATION | 24 | 48 | 36 | 45 | 31 | 65 | 97 | 62 |
| X | 34 | 59 | 39 | 69 | 36 | 50 | 75 | 36 |
| Y | 33 | 63 | 39 | 67 | 35 | 48 | 74 | 34 |

Table 6: Informativeness and Completeness per object property for all models trained with full loss on CLEVR6.

| | COMPLETENESS (↑) | | | INFORMATIVENESS (↓) | | |
| | GENESIS | MONET (ATT.) | MONET (REC.) | GENESIS | MONET (ATT.) | MONET (REC.) |
|---|---|---|---|---|---|---|
| R CHANNEL | 54 | 47 | 47 | 47 | 57 | 61 |
| G CHANNEL | 59 | 50 | 46 | 41 | 58 | 62 |
| B CHANNEL | 61 | 54 | 46 | 38 | 47 | 58 |
| SHAPE | 27 | 40 | 39 | 64 | 66 | 66 |
| MATERIAL | 56 | 56 | 53 | 28 | 39 | 41 |
| SIZE | 49 | 64 | 63 | 24 | 28 | 22 |
| ROTATION | NA | 36 | 36 | 106 | 107 | 107 |
| X | 27 | 52 | 54 | 38 | 47 | 41 |
| Y | 31 | 61 | 65 | 30 | 39 | 32 |

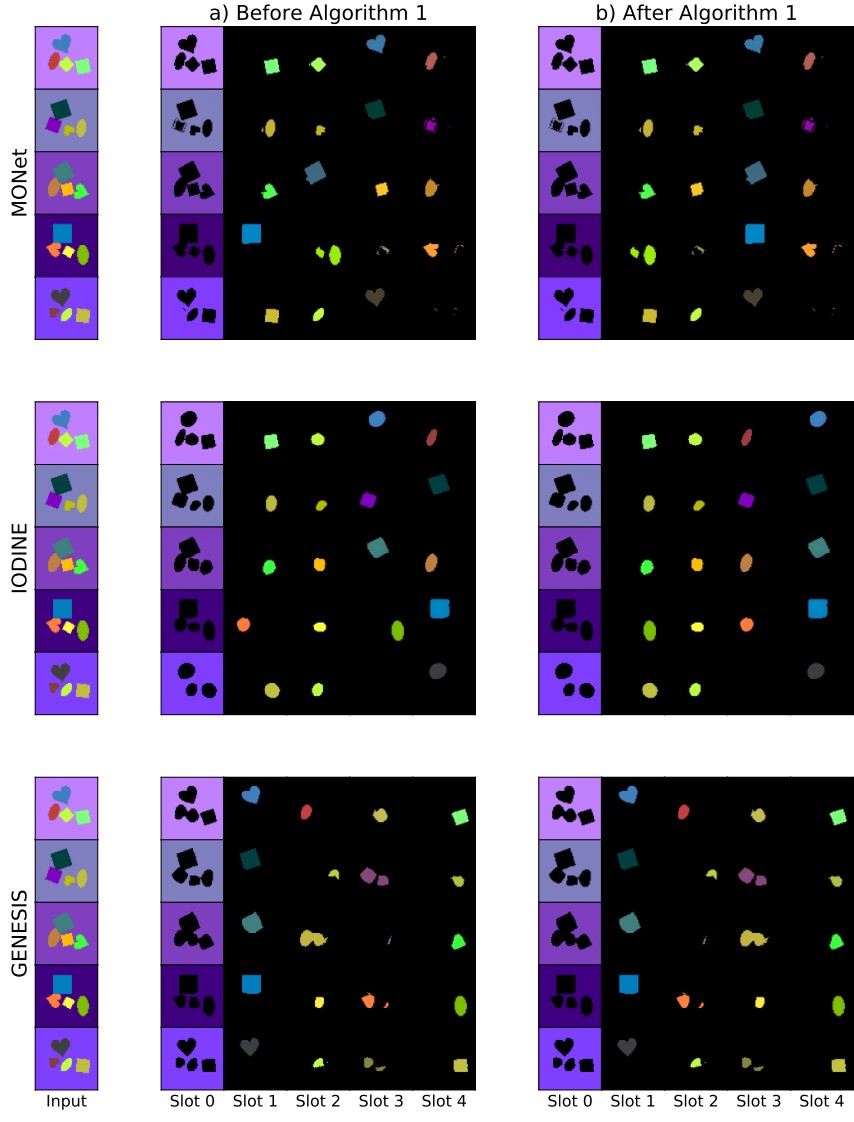

Figure 7: Visualization of the effect of Algorithm 1 on slot ordering for a group of similar inputs.

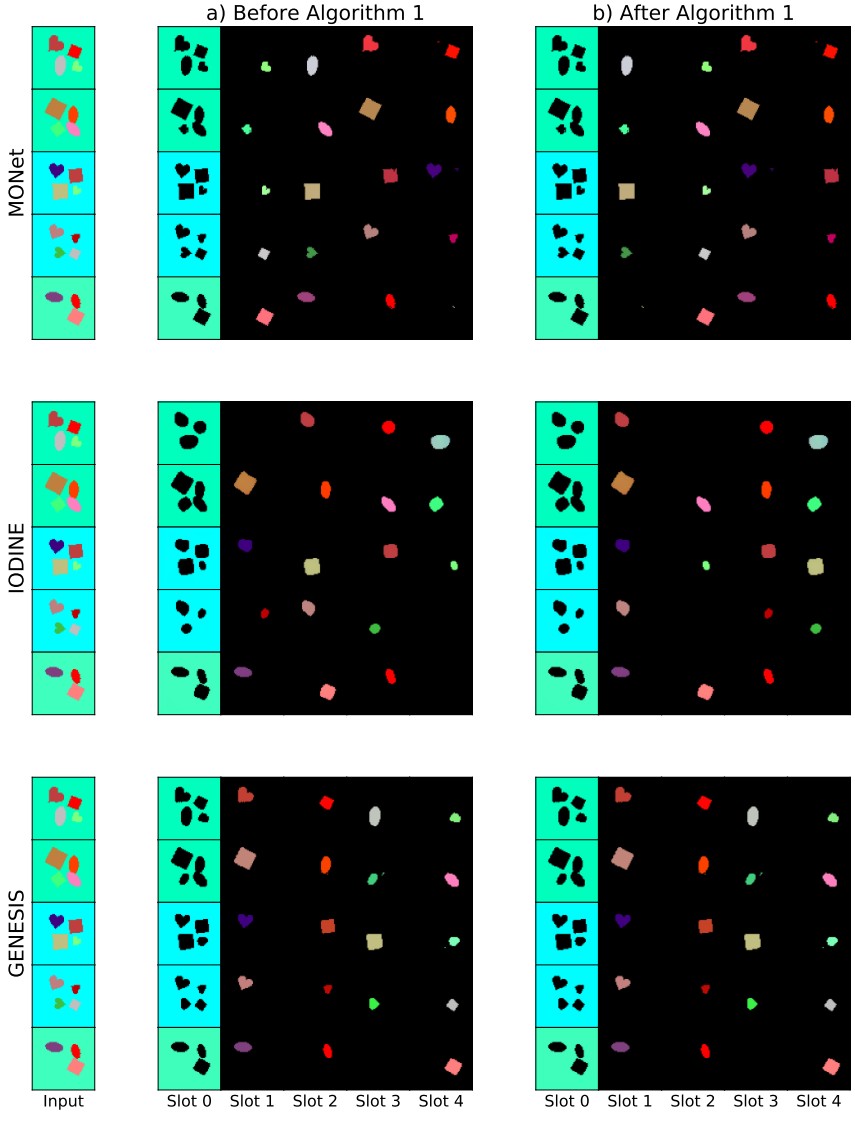

Figure 8: Visualization of the effect of Algorithm 1 on slot ordering for a group of similar inputs.

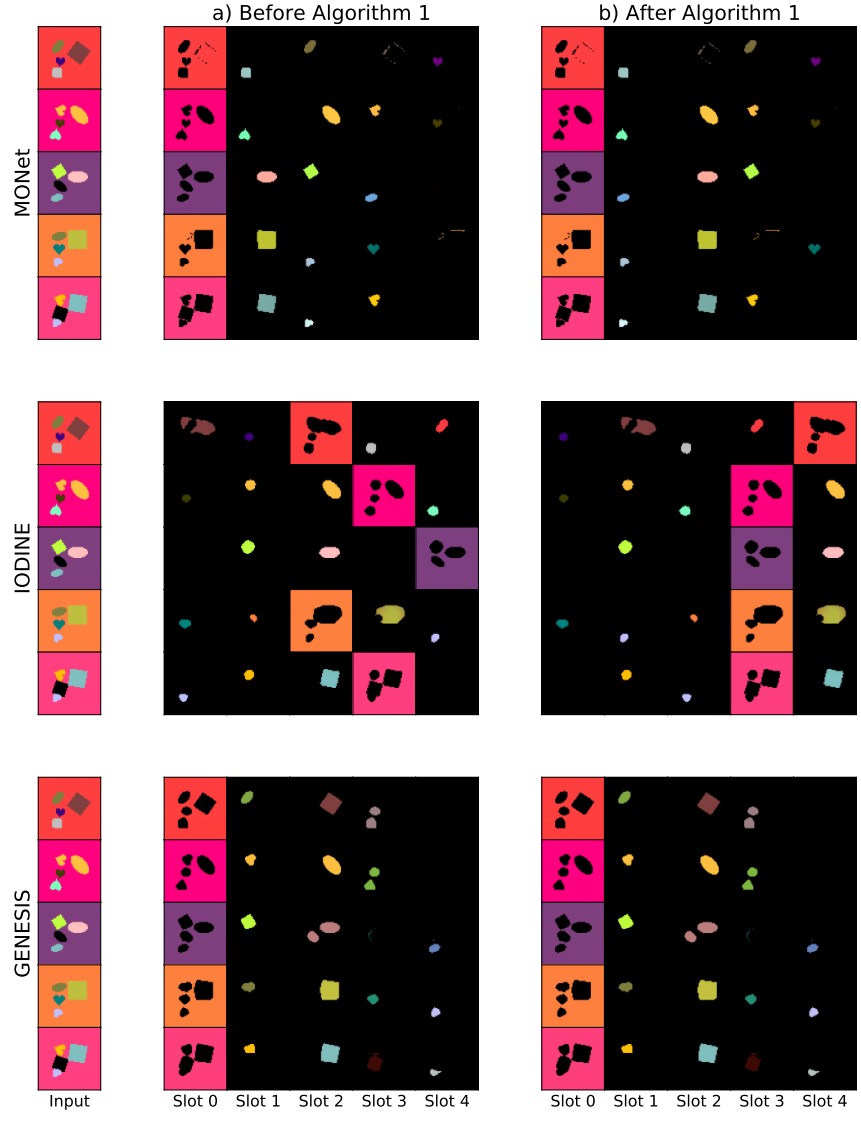

Figure 9: Visualization of the effect of Algorithm 1 on slot ordering for a group of similar inputs.

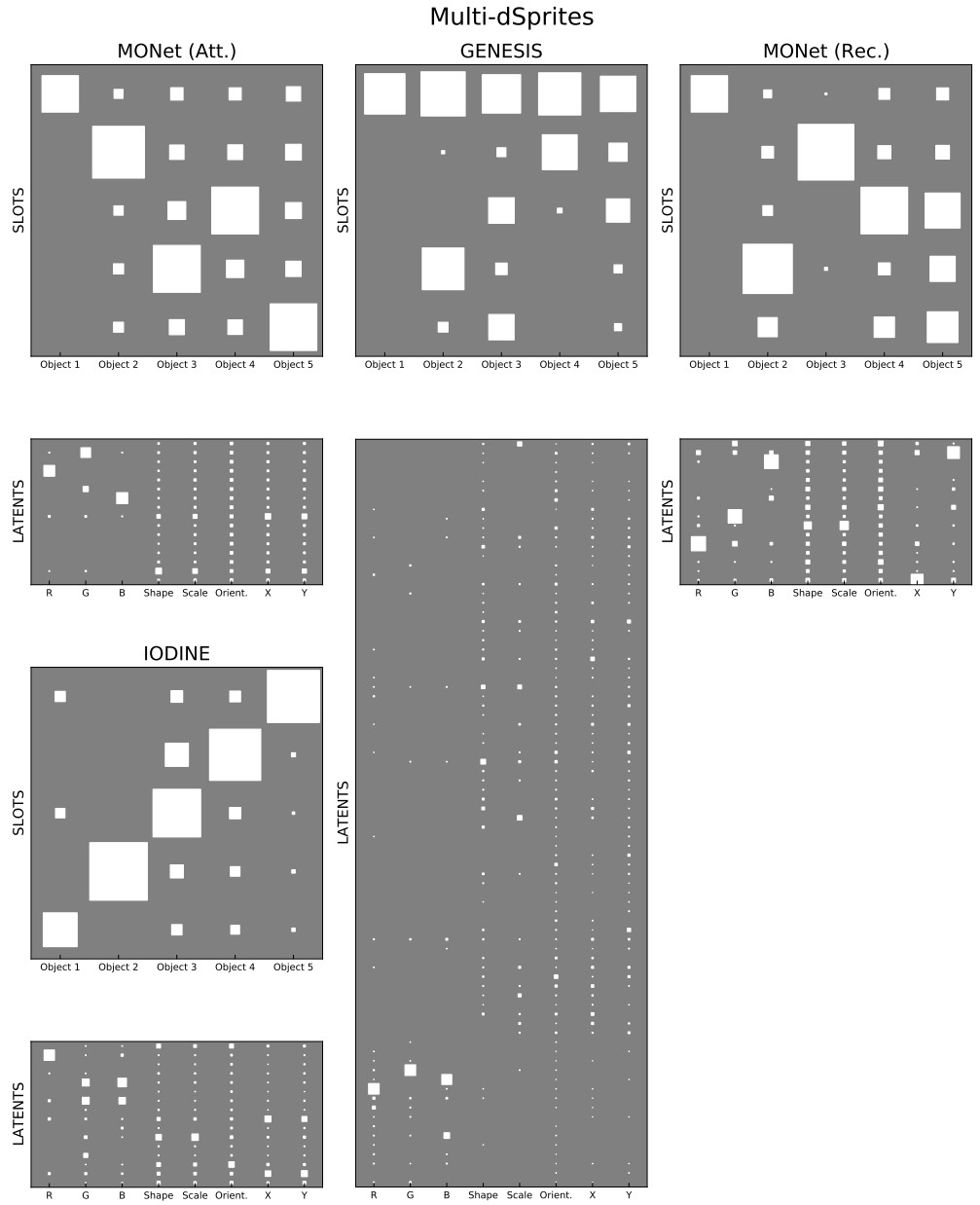

Figure 10: Projections of the affinity matrix at object-level and property-level for all four models trained with full loss on Multi-dSprites. (Hinton diagram)

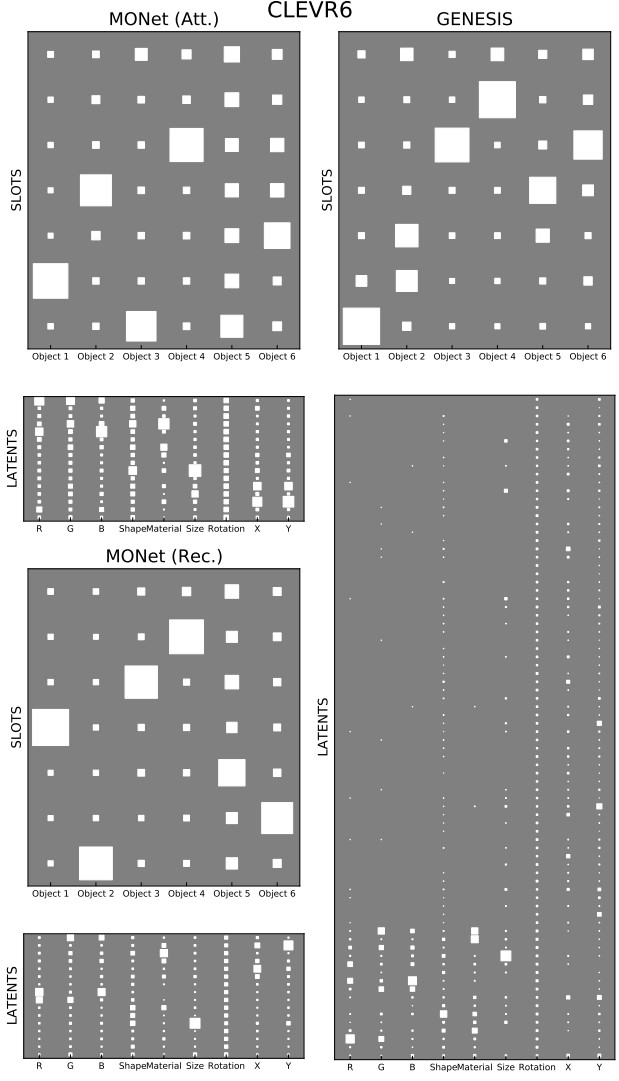

Figure 11: Projections of the affinity matrix at object-level and property-level for all three models trained with fullloss on CLEVR6 (Hinton diagram).

Table 7: Description of the factors of variation for each individual object in both datasets. For each factor, we give the set of possible values, as well as the exact locality constraints used in Algorithm 2. $x$ denotes the initial value sampled in Algorithm 2. When $x$ has values in an array, we denote $i$ the index such that $x = a[i]$. The Table indicates whether each of the considered factors is intrinsic or extrinsic. The question of whether to consider object size as intrinsic can be debated, however this does not change the results of our experiments.

| DATASET | FACTORS OF VARIATION | TYPE | POSSIBLE VALUES | LOCAL CONSTRAINT (ALG. 2) |
|---|---|---|---|---|
| MDSPR. | COLOR (R CHANNEL) | INTRINSIC | $[0, 63, 127, 191, 255]$ | $[a[i-1], a[i], a[i+1]]$ |
| | COLOR (G CHANNEL) | INTRINSIC | $[0, 63, 127, 191, 255]$ | $[a[i-1], a[i], a[i+1]]$ |
| | COLOR (B CHANNEL) | INTRINSIC | $[0, 63, 127, 191, 255]$ | $[a[i-1], a[i], a[i+1]]$ |
| | SHAPE | INTRINSIC | $\{$circle, square, heart$\}$ | $\{$circle, square, heart$\}$ |
| | SCALE | INTRINSIC | $[0, 1, 2, 3, 4, 5]$ | $[a[i-1], a[i], a[i+1]]$ |
| | ORIENTATION | EXTRINSIC | $[0, 1, \ldots, 30, 39]$ | $[a[i-3], a[i-2], \ldots, a[i+2], a[i+3]]$ |
| | X COORD. | EXTRINSIC | $[0, 1, \ldots, 30, 31]$ | $[a[i-2], \ldots, a[i+2]]$ |
| | Y COORD. | EXTRINSIC | $[0, 1, \ldots, 30, 31]$ | $[a[i-2], \ldots, a[i+2]]$ |
| CLEVR6 | COLOR (R CHANNEL) | INTRINSIC | $[0, 0.2, 0, 4, 0.6, 0.8, 1]$ | $[a[i-1], a[i], a[i+1]]$ |
| | COLOR (G CHANNEL) | INTRINSIC | $[0, 0.2, 0, 4, 0.6, 0.8, 1]$ | $[a[i-1], a[i], a[i+1]]$ |
| | COLOR (B CHANNEL) | INTRINSIC | $[0, 0.2, 0, 4, 0.6, 0.8, 1]$ | $[a[i-1], a[i], a[i+1]]$ |
| | SHAPE | INTRINSIC | $\{$cube, sphere, cylinder$\}$ | $\{$cube, sphere, cylinder$\}$ |
| | MATERIAL | INTRINSIC | $\{$rubber, metal$\}$ | $\{$rubber, metal$\}$ |
| | SIZE | INTRINSIC | $[0.35, 0.7]$ | $[0.35, 0.7]$ |
| | ROTATION | EXTRINSIC | $[\![0, 360]\!]$ | $[\![0, 360]\!]$ |
| | X COORD. | EXTRINSIC | $[\![-3, 3]\!]$ | $[\![x - 0.7, x + 0.7]\!] \cap [\![-3, 3]\!]$ |
| | Y COORD. | EXTRINSIC | $[\![-3, 3]\!]$ | $[\![x - 0.7, x + 0.7]\!] \cap [\![-3, 3]\!]$ |

# D  ADDITIONAL EXPERIMENTAL RESULTS

## D.1  VISUALIZATION OF THE PERMUTATION INFERRED BY OUR FEATURE IMPORTANCE ALGORITHM

In this Section, we inspect the slot ordering inferred by Algorithm 1, by visualizing the masked reconstruction for each slot. Note that Algorithm 1 does not make use of these reconstructions. In Figures 7,8 and 9 we compare slot ordering before (on the left) and after (on the right) applying the algorithm, for two groups of similar images (generated following Algorithm 2). We include all the considered architectures, trained on Multi-dSprites with variational loss.

On the left, we observe that the slot ordering inferred by the models is not consistent across similar images. This effect is particularly visible for IODINE, despite the fact that we set the model internal noise to a deterministic value (following observations in Greff et al. (2019)). The background slot is not even deterministic. This unpredictable ordering is extremely detrimental to the accuracy of our metric. Applying our metric to IODINE with this initial ordering yielded extremely poor values which do not do justice to the good representation learned by this model.

On the right, we see that Algorithm 1 successfully learns a consistent slot ordering, and puts matching objects at the same position. We notice almost perfect alignment, except for some failure cases with IODINE, where the background slot is switched with a missing object. We are uncertain as to the reasons for this failure, but it has limited impact on final performance as we remove the background factors in the object-level projection of the affinity matrix.

## D.2  VISUALIZATIONS OF THE LATENT SPACE

In Figures 10 and 11, we show visualizations of the projected latent space for the different architectures (trained with variational loss) for a group of inputs in both evaluation datasets. The qualitative inspection of the object-level projection is consistent with the numerical comparison: GENESIS is visually more disentangled, while MONet and IODINE achieve inferior but still satisfying disentanglement. On Multi-dSprites, the under-performance of the MONet (Att.) variant is also visible. On Multi-dSprites, we observe that the GENESIS background slot also contains information about all

objects. This is not surprising as the background mask is in some sense a negative of the different object masks. However, this phenomenon is less marked with other architectures. We think that the specific mask encoding of GENESIS somehow strengthens this duplication of information between background and object slot. Note that this does not harm performance as we removed the background slot in the final metric. We believe that this post-processing is fair because of the expected duplication effect described above.

Qualitative comparison of the property-level projections is harder due to the higher number of dimensions. Still, some observations can be made. Among object properties, color consistently obtain the best separation in the representation. On the GENESIS model, we also notice a clear separation between the last 16 dimensions that correspond to the component latent and the rest of the slot. Color is almost exclusively encoded by the component latent.

### D.3 DECOMPOSITION OF THE INFORMATIVENESS AND COMPLETENESS PER FACTOR

In Table 5, we decompose informativeness of the different models per property. Note that the completeness results do not necessarily average to the global object-level completeness as our metric involves a weighted average. On Multi-dSprites, we observe that the superior results of GENESIS mostly comes from the group of extrinsic factors that are related to the segmentation mask. This would support the hypothesis that the innovative mask encoding of GENESIS is at least partly responsible for the performance increase. However, this effect is less clear on CLEVR6.

On CLEVR6, we notice particularly bad metrics for the rotation property. This is not surprising as the rotation parameter is completely useless for two of the three shapes (sphere and cylinder) and redundant for the last one (cube). Because of this bad identifiability and affinity scores thresholding, we can not compute rotation completeness for GENESIS. This does not impact the final metric due to the weighting scheme of our metric. Concerning color channels, we note very different behavior between Multi-dSprites and CLEVR6. We believe that this is due to a particularity of CLEVR6 which is that the set of color is restricted in the training dataset.

### D.4 ABLATION OF ALGORITHMS 1 AND 2

In Table 4, we compare the object-level values given by our metric with and without Algorithms 1 and 2. We observe a significant performance drop when abalating these algorithms. This is consistent with visual inspection of the slot ordering.

