# OpenReview forum: "Evaluating Disentanglement of Structured Representations"
_ICLR.cc/2022/Conference — ICLR 2022 Poster_

### Official Review · Reviewer_ZLzf · 2021-10-25

**Correctness:** 3
**Technical Novelty And Significance:** 4
**Empirical Novelty And Significance:** 2
**Recommendation:** 6
**Confidence:** 5

**Main Review:**

**Strengths**: Valuable generalization of the DCI metrics of Eastwood et al. (2018) to object-centric representations. Both contributions (the projections and the object-matching algorithm) are novel, interesting, and of use to the object-centric representation-learning community.

**Weaknesses**: The clarity of writing, framing of the claims, and experimental evaluation are weak. These need to be improved in order to determine the strength of the proposed metrics. See points below.

**Clarity**: While Fig. 2 and Alg 1. nicely convey the main contributions, the clarity of writing could be improved as the authors tend to: i) dump in results and expect the reader to analyse them themselves (rather than taking the reader through the results with good explanations); and ii) often make statements without direct pointers to evidence to back up the statements. E.g.:
- i) *Figures 4 and 5:* what are we supposed to take from these?
- ii) *Sec. 5.3:* many statements made without evidence. E.g. why are the results "satisfying" in Section 5.3 -- disentanglement, completeness and informativeness seem poor? Where can we see the “visually satisfying results”?

**Claims/framing**: *“stronger guarantees […] than previous disentanglement metrics”*. Theorem 1 shows that DCI is a special case of the proposed metric, while Theorem 2 shows that when the proposed metric is perfect then so too is DCI. If DCI is also perfect, then surely these guarantees also apply to DCI? How are your guarantees *stronger*?

**Experimental evaluation**:
- *Representation-probing alg.*: Figure 5 seems to be the only evaluation, and it is not clear how to read this figure -- what is a “consistent slot ordering”? A better evaluation would really help here, e.g.:
    - Compare with the matchings obtained from segmentation masks.
    - Show the standard error over seeds *for the same representations/trained method* (to decouple the methods’ variability from the metric’s variability). The authors note that their framework exhibits higher variation than existing metrics -- do they attribute this to the probing algorithm? Or to their metric better-evaluating unstable methods? It would be nice to isolate these sources of variation to truly evaluate the proposed framework.

- *Disentanglement vs. informativeness*: Figure 3 and Table 2 tell a consistent story: models trained with a regularization loss that encourages disentanglement do indeed achieve better disentanglement scores with the proposed metric. However, as demonstrated in [1, Fig 6, right], [2, Table 1b], and this paper (Figure 3c), disentanglement regularization often comes at the cost of reconstruction/informativeness. Why is it then that, in Table 2, informativeness is consistently better for models *with* the disentanglement regularization? This seems very counter-intuitive as the unregularized model should capture at least as much information about the underlying factors, and raises questions about the results obtained.
- *Predictor f*: What was used in the experiments -- a linear or nonlinear predictor? E.g. lasso[2], random forests[2], or gradient-boosted trees[3]? This is very important information and seems to be missing from the main paper.

**Additional comments/questions**:
- Slot symmetry (Section 3.1, point 3): is this ever explicitly evaluated (i.e. separately to slot disentanglement)? The authors claim all 3 criteria can be evaluated by their metric -- do they mean implicitly (i.e. if slot symmetry is poor, so too will be their metric scores)?
- [4] also evaluate intra-object disentanglement (see Section 5.3) by first matching latent slots to GT objects using IoU (see Section 5.1). Your probing algorithm is more general and elegant, but it is worth mentioning this more manual approach to intra-object disentanglement in e.g. the related work section.
- “The variational loss” (Section 5.2) is quite cryptic -- I’d suggest something like “disentanglement regularization”
- Line 165: did the authors mean permutations over latent *slots* rather than latent *codes*?




[1] Higgins, I., Matthey, L., Pal, A., Burgess, C., Glorot, X., Botvinick, M., Mohamed, S., and Lerchner, A. (2017). \beta-VAE: Learning basic visual concepts with a constrained variational framework. In *International Conference on Learning Representations*.

[2] Eastwood, C. and Williams, C. K. I. (2018). A framework for the quantitative evaluation of disentangled representations. In *International Conference on Learning Representations*.

[3] Locatello, F., Bauer, S., Lucic, M., Raetsch, G., Gelly, S., Schölkopf, B., and Bachem, O. (2020). A sober look at the unsupervised learning of disentangled representations and their evaluation. *Journal of Machine Learning Research*, 21(209):1–62.

[4] Li, N., Eastwood, C. and Fisher, R. (2020). Learning object-centric representations of multi-object scenes from multiple views. In *Advances in Neural Information Processing Systems*.



**Summary Of The Paper:**

This paper proposes a new method for evaluating the disentanglement of object-centric representations. It can be seen as generalization of the DCI metrics of Eastwood et al. (2018) to object-centric representations, permitting the evaluation of both inter- and intra-object disentanglement. The two main contributions are: (i) proposing projection/marginalization operations on the matrix R in order to select an abstraction level (e.g. object or property level, as nicely illustrated in Figure 2); and (ii) an EM-based object-matching or "representation probing" algorithm which matches latent object slots with underlying GT objects (specifically, it optimizes the latent slot permutation to maximize the accuracy of the linear predictor f:z → v).

**Summary Of The Review:**

Interesting and novel ideas, but the often unclear writing, questionable claims/framing and weak experimental evaluation leave me unconvinced of their efficacy. I believe this paper is currently just below the acceptance threshold, but am eager to improve my score if the authors address the aforementioned concerns.

---

> ### Author Response · Authors · 2021-11-21
> **Response to Reviewer 3 - Clarity and Claims**
>
> We thank the reviewer for this detailed feedback. For more readability, we divided our answers into several comments.
>
> **Clarity:**
> - **Figure 5:** We extended the caption of Figure 5 to be clearer about the meaning of *“consistent slot ordering”* and the utility of the figure.
> - **Section 5.3 and Figure 4**: We modified Section 5.3 to (i) properly refer to Figure 4 for qualitative analysis of disentanglement through visual inspection of the projections of R. (ii) be more clear about which conclusions we draw from our quantitative results in Table 2. We would kindly ask the reviewer to have another look at this part.
> the *“visually satisfying results”* can be seen in Figures 5, 7, 8, 9.
> - **MONet, GENESIS and IODINE can all achieve satisfying performance with the right configuration:** By right configuration, we mean using the full loss, and with decoded masks for MONet. These models obtain between 70% and 90% object-level values for MdSprites, which is very good for two reasons
>     - The matrix R is obtained by probing representations with 80 to 480 latent dimensions. We expect this large-scale probing to be noisy.
>     - The logarithm in the conditional entropy tends to shift the values toward 0. To give a simple reference, a perfect representation where each object is influenced by a single slot obtains a perfect score of 100%, while a representation where each object is influenced by only 2 slots out of 5 obtains $1 - log_5(2) = 57%$.
>
> **Claims:**
>
> Our claims are the following:
> - *“We theoretically show that our framework gives stronger guarantees of selecting a good model than previous disentanglement metrics.”*
> - *“one dimensional projections provide a sound substitute for prior unstructured metrics”*
>
> To support these claims, our theorems show that when our metric is good ($\ge 1 - \epsilon$, where 1 is a perfect score), then DCI is also good ($\ge 1 - 2\epsilon$). To make an analogy, suppose that there exists two standardized tests A (ours) and B (DCI) for university admission. Suppose that we can formally prove that B is easier than A, in the sense that a candidate who passed A is sure to pass B. Then we can say that a good grade for test A gives stronger guarantees of a candidate's ability than a good grade for test B (and not the opposite, because a candidate who passed B might still fail A. We believe that this point is the source of the reviewer's confusion.). Therefore, a selective university can decide to only rely on test A, especially if test A gives additional guarantees (in our case, object-level disentanglement, which is not measured by DCI)

---

> > ### Comment · Reviewer_ZLzf · 2021-11-26
> > **Response to authors**
> >
> > **Clarity:**
> > - These clarifications will improve the paper, and I hope the explanation of the "right" configuration and explicit pointers to the "visually satisfying results" will be included in the paper.
> >
> > **Claims:**
> > - $\geq 1 - \epsilon$ vs. $\geq 1 - 2\epsilon$: I must have missed this comparison, or perhaps I misunderstood your theorem. Are you saying that your metric has a higher lower-bound on dis/compl. here vs. DCI? Where K=2 is the number of objects? And this assumes correct object matching?
> > - *Analogy*: For this analogy to work, we would need to trust imperfect scores on test A, i.e. be sure that the things that make it more difficult than test B correctly assess the candidates ability. Otherwise, if we pass test B but not test A, this could simply be a flaw of test A (incorrectly assesses the candidates ability to give an imperfect score when it should be perfect). Do you provide such guarantees for the correctness of **imperfect scores** with your metric (test A)? Perhaps I have missed this guarantee, and it is the root of my confusion here.

---

> > > ### Author Response · Authors · 2021-11-28
> > > **Response to Reviewer 3 - Claims**
> > >
> > > **Claims:**
> > > - *$ \ge 1 - \epsilon $ vs. $ \ge 1 - 2\epsilon $*: this point is explained in slightly more general terms in the last paragraph of Section 3.3. $k = 2$ is the number of hierarchy levels in the representation. To rephrase this again: our results show that when both object level and property level disentanglement are good ($ \ge 1 - \epsilon $), then flat DCI disentanglement is also good  ($ \ge 1 - 2\epsilon $). This makes no assumption of correct object matching.
> > > - **Do you provide such guarantees for the correctness of imperfect scores with your metric (test A)?** The theoretical claims in the main paper are focused on good scores
> > >     -  *“We theoretically show that our framework gives stronger guarantees of selecting a good model than previous disentanglement metrics.”* Here it is implicitly meant *"when our metric scores are good"* (which is always the case when selecting a good model).
> > >     - *“one dimensional projections provide a sound substitute for prior unstructured metrics”* here, we deliberately used the notion of soundness (by opposition to completeness). In mathematical logic, a soundness result means that we can not select (or prove) something that is bad (or wrong). On the contrary, a completeness result means that we can not reject or (disprove) something that is good (or true).
> > > - **Completeness results for our framework (related to imperfect scores)**
> > >      - For high-level understanding, note that by definition (see the presentation of our framework in Section 3.1), an imperfect score for our metrics (test A) either means that (i) object-level disentanglement is bad (ii) properties are not correctly disentangled inside slots (iii) slots are not symmetric. All these three criteria are without doubt desirable.
> > >      - For a theoretical analysis of imperfect scores, we refer to Theorems 3 and 4 in the appendix. These results show that it is possible to obtain a bad score for our metric (test A) while obtaining a good score for DCI (test B), when the object-level projection and the property-level projection have a spurious correlation, quantified by the conditional mutual information terms $A_s$ and $B_s$. These theorems are a core part of our work, but they provoked some confusion in early versions of our work, because of the difficulty to interpret the terms $A_s$ ans $B_s$. This is why we decided to reserve it to interested readers.

---

> > > > ### Comment · Reviewer_ZLzf · 2021-11-28
> > > > **Response to authors**
> > > >
> > > > I appreciate the authors response and continued engagement. I now understand the claim, and have increased my score to 6 as a result. However, I do think that the claim is missing a condition since, for "flat" representations, the guarantees are the same as previous metrics. E.g.:
> > > > - *"We theoretically show that our framework gives stronger guarantees of selecting a good model than previous disentanglement metrics **[for object-centric representations]**"*

---

> > > > > ### Author Response · Authors · 2021-11-29
> > > > > **Response to Reviewer 3**
> > > > >
> > > > > Thanks for increasing your score. We will adapt this claim in future versions

---

> ### Author Response · Authors · 2021-11-21
> **Response to Reviewer 3 - Experimental Evaluation and Additional**
>
> We thank the reviewer for this detailed feedback. For more readability, we divided our answers into several comments.
>
> **Experimental evaluation:**
> - **Evaluation of representation-probing:**
>     - we performed a new experiment where we compared the permutation learned by our method with the segmentation masks based IoU matching. We added our results in Table 1 on page 7. They show that on average, both methods return the same permutation on 62\% of inputs, which is several orders of magnitude better than chance level of 0.83% (because there are 120 possible permutations).
>     - we added a new plot to verify convergence of the EM probing algorithm in less than 20 iterations for all models. See Figure 6 on page 7.
> - **Sources of variability:** for all architectures, we performed different runs of the metric on the same trained models and observed that the standard deviation of our metrics is always below 1\%, indicating that our metric has little variability across runs on the same trained model, and than our framework better-evaluates unstable methods.
>
> - **Disentanglement vs Informativeness**: we would like to clarify some aspects about the results mentioned by the reviewer
>     - [1, Fig 6, right] shows that disentanglement comes at the cost of reconstruction quality. There is no mention of informativeness in this figure. Reconstruction quality and informativeness are not necessarily always correlated, because informativeness is not influenced by segmentation mask sharpness.
>     - [2, Table 1b] (random forest) indeed shows a disentanglement - informativeness tradeoff, but [2, Table 1a] (Lasso) shows that better disentanglement leads to better informativeness. The explanation given in [2] is the following: *“the informativeness metric has some overlap with the disentanglement metric, with the size of the overlap determined by the capacity of f”*. For the single object datasets in [2], the capacity of random forests is large enough to have no overlap. But for our more complex multi-object datasets, there can be an overlap between disentanglement and informativeness.
> - **Predictor f:** As stated in the supplementary, all the final metrics are computed with random forests, similarly to [2]. We mentioned this in the main paper.
>
> **Additional:**
> - Slot symmetry and slot disentanglement are indeed evaluated together in a single metric. We have clarified this in the paper (Section 3.1, *“with a single metric”*).
> - We have mentioned the IoU matching of [4] in the related work (Appendix A.3). Note that the 1-1 matching given by IoU is not sufficient for our method, as we need all-pairs scores to measure row and column entropy (see the answer to reviewer 2 about Hungarian matching).
> - We changed *“ablation of variational loss”* to *“ablation of disentanglement regularization”* throughout the paper.
> - We chose to not write *“permutation over latent slots”* because our approach is more general and adapts to any permutation of groups of latent dimensions. For instance it would be possible to allow for permutation of the mask and component latents. We changed to *“permutations over groups of latent dimensions (for instance slots)”* Does the reviewer find this new formulation adapted?

---

> > ### Comment · Reviewer_ZLzf · 2021-11-26
> > **Response to authors**
> >
> > Dear authors,
> >
> > Thank you for your response and for including the additional experiments, analysis, clarifications, and updated descriptions. I believe they improve the paper.
> >
> > I am satisfied with all the above points, except for the one on **Disentanglement vs Informativeness**.
> > - *Reconstruction quality*: [1] do indeed show a trade-off between disentanglement and reconstruction, rather than informativeness. However, I disagree with your subsequent point as an explanation for why we don't see a trade-off between disentanglement and informativeness in your paper---informativeness and reconstruction do tend to strongly correlate if the predictor $f$ has been properly trained (more on this below).
> > - *Effect of $f$'s capacity on the informativeness-disentanglement trade-off*: Indeed [2, Table 1] shows a disentanglement-informativeness trade-off for the nonlinear random forest predictor, but not for the linear Lasso predictor. As explained in [2], this is because disentangled representations tend to be more "explicit", i.e. better predict the ground-truth factors with restricted capacity. However, you are using the random forest predictor here so *should* see a trade-off. I do not agree with your claim that random forests, as a model class, do not have enough capacity to map between the representation and ground-truth factors for multi-object datasets. As in [2], choosing the hyperparameters of the predictor $f$ using a validation set is critical, e.g. for random forests, choosing a max tree depth that gives you enough capacity for your dataset. If this is done properly, as [2] stress it needs to be, then the overlap should be diminished and one should see a trade-off between disentanglement and informativeness.
> > - *Questions*:
> >   1. Did you fit the hyperparameters of $f$ on a validation set? Not doing may explain the lack of a trade-off here. In Appendix C.4, I see *"For the final predictor, we use a random forest with 10 trees, and a maximum depth of 15"*. [2] choose different max tree depths for each ground-truth factor, based on a validation set.
> >   2. Perhaps reporting the reconstruction errors too would clear things up? I expect we would then see the inherent trade-off, which would focus this line-of-inquiry on the fitting of $f$ (in particular, the informativeness).

---

> > > ### Author Response · Authors · 2021-11-28
> > > **Response to Reviewer 3 - Disentanglement vs Informativeness**
> > >
> > > We understand the reviewer’s interrogations but we are confident in our experimental methodology. To answer each point in detail
> > > - **Did you fit the hyperparameters of f on a validation set?** Yes, initially we were comparing different max tree depths (5, 10, 15, 20, 25, 30) on a validation set . But we observed that (i) a value of 15 generally gives better validation performance than 5 and 10 (ii) values > 15 do not significantly improve validation performance and get very slow to fit. This is why we chose to always go with 15. Note that this optimal value might be related to the number of samples used to fit the predictor (currently 4000).
> > > - **I do not agree with your claim that random forests, as a model class, do not have enough capacity to map between the representation and ground-truth factors for multi-object datasets.** Based on our observation that values > 15 do not significantly improve validation performance, we unfortunately have to disagree with the reviewer on this point. But this might also depend on the number of samples used to fit the predictor.
> > > - **Disentangled representations tend to be more "explicit"** If the reviewer finds it more adapted, we are open to replacing all mentions of *informativeness* by *explicitness* in the paper.
> > > - **Perhaps reporting the reconstruction errors too would clear things up?** For Multi-dSprites, the sum of squares reconstruction error is the following (lower is better). We observe the expected tradeoff between disentanglement and reconstruction error, except for Iodine, where the values are similar. We can add this information to the paper.
> > >
> > > Model           | Full loss | Without disentanglement regularization
> > >
> > > Monet (Att)   |        160 |                                       42
> > >
> > > Monet (Dec) |          53 |                                       33
> > >
> > > Genesis        |         36 |                                         6
> > >
> > > Iodine           |          39 |                                       39

---

> > > > ### Comment · Reviewer_ZLzf · 2021-11-28
> > > > **Response to authors**
> > > >
> > > > I am happy that the authors are confident in their experimental methodology. However, for a reader to share this confidence, details on the how the hyperparameters were selected need to be reported -- they are of critical importance to this framework, directly controlling the trade-off between disentanglement and informativeness.
> > > >
> > > > We shall agree to disagree on the importance of fitting tree-depth to a validation set on a per-factor basis. I believe that this likely explains the lack of visible trade-off between disentanglement and informativeness, and do not think that values > 15 should be ignored because they result in a slow fit or do not *significantly* improve performance. Such a choice biases the scores to be more disentangled and less informative. Having said this, unbiased scores are more important when evaluating specific methods than illustrating the efficacy of new measures, as is being done here. Thus, I am happy enough that some rough fitting was done using a validation set, and hope the authors include the above details in the Appendix. As a result, I increase my score to 6.
> > > >
> > > > I think reporting the reconstructions errors would be helpful, but do not think that informativeness should be renamed explicitness -- explicitness is not what is being measured by the NRMSE, it is indeed informativeness.

---

### Official Review · Reviewer_uWyU · 2021-10-31

**Correctness:** 3
**Technical Novelty And Significance:** 3
**Empirical Novelty And Significance:** 3
**Recommendation:** 6
**Confidence:** 3

**Main Review:**

This paper is well-written and of overall high quality. The problem addressed in this paper -- evaluation of object-centric generative models -- is of high relevance to the field, especially given the limited choice of metrics available at present and the unsuitability of standard disentanglement metrics for multi-object scenes.

The metric introduced in this paper is simple, novel, and of high significance. The experiments are insightful and allow for discussion/evaluation of prior work under a new light -- in the context of feature disentanglement. Especially the insight that regularizers on the latent space (as used in variational objectives) appear to have a positive impact on disentanglement scores but a negative impact on (visual) scene decomposition scores (e.g. segmentation mask ARI) is interesting and will likely have an impact on future work in this area.

In its present form, I still have two major concerns with the paper:

* I am unsure about the significance of the core technical contribution of this work (i.e., the representation probing algorithm to handle slot permutation invariance). It is unclear to me why earlier approaches, such as training a per-slot probe using Hungarian matching w.r.t. a set of target variables (as done in Slot Attention; Locatello et al., NeurIPS 2020), are unfit for this task. Appendix A.2 explains that the probing/matching approach used in Slot Attention does "not permit to identify which latent slot was used for the prediction, which is essential to measure disentanglement". This statement is unclear to me, as the Hungarian matching algorithm returns a permutation matrix which does indeed provide 1-to-1 correspondence between slots and targets, i.e. we can directly read off which slot was used for the prediction. It is possible that I misunderstand the argument by the authors, and I would appreciate it if this could be clarified.

* The images in Figure 1 appear to have been taken/copied from Figure 20 in the Slot Attention paper. Copying parts of figures without referencing the original work is not good scholarship and should be fixed. Since these are visualizations of model predictions, it should be fine to include them here without asking for explicit permission by the authors, but the source should at least be acknowledged in the figure caption.

Other comments:
* It would be good to include y-axis labels in Figure 4 (or a more descriptive annotation). In its current form, it is unclear whether the x- or the y-axis represents latent slots (vs. ground truth objects).
* This sentence is (grammatically) unclear (Section 3): "[...] since it provides stronger guarantees than the selected model is correctly disentangled".


**Summary Of The Paper:**

This paper introduces a new metric for evaluating disentanglement of latent representations, which -- unlike prior work -- supports object-centric, structured latent spaces with a set of latent variables (as opposed to a single global latent variable). While prior work on object-centric generative models primarily evaluated their ability to decompose visual scenes into spatial regions corresponding to individual objects, this novel metric allows for systematic evaluation of disentanglement of learned object properties within and across "object slots" (i.e. individual latent variables) in these models. The paper further introduces a representation probing algorithm using an EM-style iterative procedure, which is invariant to slot permutation. The metric is experimentally demonstrated on standard unsupervised scene decomposition datasets and models.

**Summary Of The Review:**

Overall, this paper presents a significant advance in evaluation of object-centric generative models and provides a sound new metric for this task. Despite my concerns mentioned above, I believe this paper can be accepted (although it is borderline).

---

> ### Author Response · Authors · 2021-11-21
> **Response to Reviewer 2**
>
> We thank the reviewer for these remarks. We have made the following modifications
> - **Comparison with Hungarian matching:** We agree with the reviewer that the Hungarian matching algorithm returns a permutation which provides 1-to-1 correspondence between slots and targets. However, this permutation only indicates the most relevant slot for each target. In contrast, our EM-probing method computes an importance score for each slot/target pair, which is needed to measure row and column entropy. Our statement *“the matching approach used in Slot Attention does not permit to identify which latent slot was used for the prediction"* was not accurate, and we changed to *“the matching approach used in Slot Attention only identifies the most important slot for each target, but our framework requires an importance score for each slot/target pair"* in appendix A.3.
> - **Images in Figure 1:** we changed to visualizations of one of our Genesis trained models.
> - **Figure 4:** the x-axis represents objects/properties, while the y-axis represents slots/latents. We changed the y-axis labels to better reflect this.
> - **Typo:** we corrected "guarantees than the selected model is correctly disentangled" into “guarantees **that** the selected model is correctly disentangled”.

---

> > ### Comment · Reviewer_uWyU · 2021-11-22
> > **Re: Response to Reviewer 2**
> >
> > Thanks a lot for your response and for your clarification w.r.t. Hungarian matching. This makes a lot of sense and I appreciate the clarification in the paper revision.
> >
> > As you require importance scores for each slot/target pair, did you consider using the Sinkhorn algorithm (instead of Hungarian matching) to obtain a soft matching? To me, this feels like a more natural choice, as it is a well-studied algorithm that is frequently used for matching problems. Even if you would like to keep your EM-probing method as the default, it would be very insightful for the reader to have a comparison against a more canonical matching method. I'd be curious to hear your thoughts on this.

---

> > > ### Author Response · Authors · 2021-11-26
> > > **Re: Response to Reviewer 2**
> > >
> > > Thanks to this suggestion we have indeed found some very related work using Sinkhorn matchings to learn latent permutations [1], based on a similar observation that *“exact marginalization over these combinatorial objects [permutations] is intractable”*. We believe that it makes sense to keep EM probing as a default because:
> > > - it can be motivated and described very concisely (see Section 4 and Algorithm 1).
> > > - it is the most lightweight approach, which only requires a few EM iterations (see Figure 6 to visualize speed of convergence).
> > > - mathematically, it is a standard but elegant way to solve intractable optimisation problems, which doesn't require differentiability.
> > >
> > > For future work, we agree that it would be very interesting to check if the more elegant and complex Gumbel-Sinkhorn networks can beat the 62% agreement between EM probing and IoU matching (reported in Table 1). We will mention this in the paper.
> > >
> > > [1] Learning latent permutations with Gumbel-Sinkhorn networks. Mena, Belanger, Linderman, Snoek, ICLR 2018.

---

> > > > ### Comment · Reviewer_uWyU · 2021-11-28
> > > > **Response to authors**
> > > >
> > > > Thank you for your clarification, I agree that it makes sense to keep your EM probing algorithm as a technical contribution.
> > > >
> > > > While I think a direct comparison to Sinkhorn-based matching would be valuable, it is not the highest priority for improving the paper -- I would instead recommend to primarily focus on the concerns and suggestions that reviewer ZLzf raised.
> > > >
> > > > Note that Gumbel-Sinkhorn networks would give you a discrete 1-to-1 matching, i.e. what you earlier described as being not very useful, since you need pairwise scores. You might instead just want to use the Sinkhorn distance directly as basis for your score.
> > > >
> > > > See e.g. [1] for a common reference on the Sinkhorn distance, and [2] for a more high-level introduction in the form of a blog post.
> > > >
> > > > [1] Cuturi, "Sinkhorn Distances: Lightspeed Computation of Optimal Transport" (N[eur]IPS 2013)
> > > > [2] Daza, "Approximating Wasserstein distances with PyTorch" (Blog post, 2019): https://dfdazac.github.io/sinkhorn.html

---

### Official Review · Reviewer_ERyE · 2021-11-04

**Correctness:** 3
**Technical Novelty And Significance:** 3
**Empirical Novelty And Significance:** 3
**Recommendation:** 6
**Confidence:** 4

**Main Review:**

Strengths:
+ Compositional disentanglement is necessary when there are multiple objects of interest in an image. This paper addresses the problem of evaluating models that learn compositional representations.
+ Permutation-invariant representation probing is a necessary contribution that extends the mapping between latents and generative factors to the object-centric case.
+ The experiments performed and results displayed are comprehensive and provide a holistic view of the metric within existing methods.

Weaknesses:
- It would be better to include some real-world examples of the need for compositional disentanglement and the motivation for more than one slot (i.e., global disentanglement).
- How does this study of compositional disentanglement address spurious correlations in the data? Especially, if object and color are spuriously correlated, can ‘slot disentanglement’ identify such spurious correlations?
- Object-level disentanglement says that there should be a one-to-one mapping between slots and objects. However, the example given in Figure 2 has two slots for each object. It would be better to explicitly state how the matrix is reduced to the mapping in empirical studies.
- Usually, it makes sense to visualize each row of affinity matrix R to sum to 1. However, in section 3.2, it is mentioned that R can be normalized to make the sum of entries of R equal to 1. Is this a valid way of normalizing R? If yes, explain it clearly as this idea is being used in proofs of theorems.
- In the experimental results, the way we find the dimensions to project on is not clear. I think this is also important in identifying the invariant permutations of latent codes.
- The paper states: “ablation tends to improve pixel-level segmentation. This would imply that visual object separation is directly caused by architectural inductive biases, which puts into question the implicit beliefs of prior work.” Isn’t this expected as the ablation performed removes regularization (needed for disentanglement) and the subsequent improvement is merely proof that the object separation works?

Minor Comments:
Theorem 1 in main paper is referred to as Theorem 3 in Appendix


**Summary Of The Paper:**

This paper proposes a metric for evaluating disentanglement in compositional representation learning upending existing global disentanglement metrics, that disregard any representational structure. The proposed metric is based on the projections of the affinity matrix proposed earlier in the DCI metric. The paper also proposes an EM-like permutation-invariant formulation for obtaining the relative importance of a latent in predicting a generative factor. This allows for slots to be permuted wlog. Empirical studies are performed to study the disentanglement of existing compositional representation learners.

**Summary Of The Review:**

This paper addresses a timely problem and the proposed evaluation metric and representation probing methods are sound. However, this paper needs to explain some points more clearly to better understand the work.

---

> ### Author Response · Authors · 2021-11-21
> **Response to Reviewer 1**
>
> We thank the reviewer for these comments. We have addressed the following points:
> - **Real-world examples of the need for compositional disentanglement:** In the first paragraph of the introduction, we added examples of concrete settings in which having a compositional representation can bring performance gains.
> - **Spurious correlations in the data:** Mainly, our framework introduces more robustness to spurious correlations. Object-level disentanglement is robust to spurious correlations of properties (such as position and color), because it marginalizes properties. Property-level disentanglement is robust to spurious correlations between objects because it marginalizes objects. But we do not claim the ability to identify spurious correlations.
> - **Clarification about Figure 2, which has two slots for each object:** There is a misunderstanding concerning this point. It is necessary to compute the affinity between slots and objects in an all-pairs manner, because we can’t know a priori which slot represents which object. Then, a mapping can be obtained by looking for the maximum of each column. We have followed the reviewer’s advice and stated this in the caption of Figure 2.
> - **Normalization of R:** As part of the proof of Theorem 1, we formally show that our global normalization of R is compatible with row-wise normalization used in the DCI framework. This is the case because the mathematical definition of conditional entropy involves a renormalization step happening after projection. This is now indicated in Figure 2 for more clarity.
> - **How we find the dimensions on which to project:** all the models in this line of work (MOnet, Genesis, Iodine, Slot Attention…) assume a predetermined slot structure, which means that all the dimensions to project on are well-identified. We found it fair to make the same assumption.
> - **Isn’t this expected (that visual object separation is directly caused by architectural inductive biases) as the ablation performed removes regularization (needed for disentanglement) and the subsequent improvement is merely proof that the object separation works?:** we would still have expected some positive impact of disentanglement regularization on object-level separation (which is a form of disentanglement). To address the remark, we have changed the statement to *“This would imply that visual object separation is **only** caused by architectural inductive biases, which clarifies the impact of disentanglement regularization in prior work”.*
> - **Theorem numbering:** this is now corrected.

---

### Author Response · Authors · 2021-11-21
**Summary of Modifications to the Submission**

**Writing/Clarity:**
- In the first paragraph of the introduction, we added examples of concrete settings in which a compositional representation can bring performance gains.
- In the caption of Figure 2, we indicated how to obtain a mapping from object to slots by taking the column maximum. We also clarified that the measure of completeness (resp. disentanglement) involves a row-wise (resp. column-wise) renormalization of R.
- Changed to *“This would imply that visual object separation is **only** caused by architectural inductive biases, which clarifies the impact of disentanglement regularization in prior work”* .
- Correction of *“guarantees than the selected model is correctly disentangled”* into *“guarantees that the selected model is correctly disentangled”* .
- Explained the limitation of the Hungarian matching approach in supplementary A.3 *“the matching approach used in Iodine and Slot Attention only identifies the most important slot for each target,  but the DCI framework requires an importance score for each slot/object pair.”*
- Changed the images in Figure 1.
- Changed the y-axis in Figure 4 .
- Changed *“ablation of variational loss”* to *“ablation of disentanglement regularization”*.
- In Section 5 (Models), we indicated that our final feature importances are obtained with random forests.
- In Section 3.1, we clarified that slot disentanglement and slot symmetry are evaluated together in a single metric.
- In Section 4., changed *“permutation of latent codes”* into *“permutations over groups of latent dimensions (for instance slots)”*.
- Added a reference to the IoU matching of MulMON in the related work about permutation-invariant learning, in Appendix A.3.
- We rewrote Section 5.3 to incorporate different remarks.
- Extended the caption of Figure 5.
**Experiments:**
- we performed a new experiment where we compared the permutation learned by our method with the segmentation masks based IoU matching. We added our results in Table 1 on page 7. They show that on average, both methods return the same permutation on 62% of inputs, which is several orders of magnitude better than chance level of 0.83% (because there are 120 possible permutations).
- we added a new plot to verify convergence of the EM probing algorithm in less than 20 iterations for all models. See Figure 6 on page 7.
- To isolate sources of variability in our metrics, we performed different runs of the metric on the same trained models and observed that the standard deviation of our metrics is always below 1%, indicating that our metric has little variability across runs on the same trained model, and than our framework better-evaluates unstable methods.

---

### Public Comment · ~Raphaël_Dang-Nhu1 · 2022-01-29
**Camera-ready version**

I have uploaded the camera-ready version with the following changes:
- switched to Times font
- changed to "We theoretically show that for structured representations, our framework gives stronger guarantees of selecting a good model than previous disentanglement metrics"
- changed title to "Evaluating disentanglement of structured representations"
- added explanations about max tree depth selection in the supplementary
- added reference to Gumbel Sinkhorn networks in the supplementary
- added the following acknowledgements

*"This work was granted access to the HPC resources of IDRIS under the allocation 2020-AD011012138 made by GENCI. We would like to thank Frederik Benzing, Kalina Petrova, Asier Mujika, and Wouter Tonnon for helpful discussions. This work constitutes the public version of Raphaël Dang-Nhu’s Master Thesis at ETH Zürich."*

---

### Decision · Program_Chairs · 2022-01-20

**Decision:**

Accept (Poster)

**Comment:**

This paper presents a new metric for disentanglement of learned representations, extending a prominent framework (DCI) to support object-centric structured representations.

The reviewers agree on the importance of the question and find the metric a valuable contribution for addressing this problem. In the discussion, the reviewers identified some clarity issues that the authors have improved, leading to an overall much better writeup, as well as some deeper evaluation of learned matching agreements. The main remaining points that could be improved are
 - making the results more robust with thorough hyperparam tuning
 - connecting to other methods for inducing soft / probabilistic matchings, such as Sinkhorn or smooth&sparse optimal transport.

Please consider switching to the Times font as recommended by the ICLR style guide.

---

> ### Public Comment · ~Raphaël_Dang-Nhu1 · 2022-01-29
> **Author's comment (public copy)**
>
> I again thank the reviewers and chairs for this constructive discussion. In response to the above summary of the contribution, I would like to emphasize that the application scope of this work goes beyond object-centric representations. In fact, the hierarchical approach presented here has the potential to apply to any kind of structured generative modeling, since it operates directly at latent space level. It can thus be of use to the representation learning community as a whole. I have chosen to focus on object-centric representations as a first application because I see it as the quintessential form of compositional structure. As a first step towards generalization, Section 5.4 demonstrates applicability to measuring disentanglement of intrinsic and extrinsic object properties, which can be naturally extended to disentanglement of shape and appearance. For future work, I plan to investigate how this can be applied to style transfer architectures.